# From Obesity to Hippocampal Neurodegeneration: Pathogenesis and Non-Pharmacological Interventions

**DOI:** 10.3390/ijms22010201

**Published:** 2020-12-28

**Authors:** Thomas Ho-yin Lee, Suk-yu Yau

**Affiliations:** Department of Rehabilitation Sciences, Faculty of Health and Social Sciences, Hong Kong Polytechnic University, Hung Hom, Hong Kong; thomas.hy.lee@connect.polyu.hk

**Keywords:** exerkines, neuroinflammation, obesity, diabetes, hippocampal plasticity, physical exercise

## Abstract

High-caloric diet and physical inactivity predispose individuals to obesity and diabetes, which are risk factors of hippocampal neurodegeneration and cognitive deficits. Along with the adipose-hippocampus crosstalk, chronically inflamed adipose tissue secretes inflammatory cytokine could trigger neuroinflammatory responses in the hippocampus, and in turn, impairs hippocampal neuroplasticity under obese and diabetic conditions. Hence, caloric restriction and physical exercise are critical non-pharmacological interventions to halt the pathogenesis from obesity to hippocampal neurodegeneration. In response to physical exercise, peripheral organs, including the adipose tissue, skeletal muscles, and liver, can secret numerous exerkines, which bring beneficial effects to metabolic and brain health. In this review, we summarized how chronic inflammation in adipose tissue could trigger neuroinflammation and hippocampal impairment, which potentially contribute to cognitive deficits in obese and diabetic conditions. We also discussed the potential mechanisms underlying the neurotrophic and neuroprotective effects of caloric restriction and physical exercise by counteracting neuroinflammation, plasticity deficits, and cognitive impairments. This review provides timely insights into how chronic metabolic disorders, like obesity, could impair brain health and cognitive functions in later life.

## 1. Introduction

Western dietary patterns and sedentary lifestyle have fueled the current obesity epidemic [1]. Processed and refined food is constituted by saturated fats, added sugar, and salts, which contribute to high caloric intake under chronic consumption. Besides, the primary consumption of red meats and dairy products with a lack of vegetables and fresh fruits are the other characteristics of the Western dietary pattern. However, when caloric consumption far exceeds expenditure under prolonged physical inactivity, metabolic syndromes are developed. Metabolic syndrome encompasses a cluster of risk factors that could lead to cardiovascular diseases and diabetes [2]. Criteria for clinical diagnosis of metabolic syndrome include increases in waist circumference, triglycerides, blood pressure, fasting glucose, and a reduction in high-density lipoprotein cholesterol [3]. A recent review has summarized that the key pro-inflammatory constituents from the dietary sources, including saturated fatty acids, cholesterol, added sugars, refined grains, purines, dietary carnitine, and dietary histidine [4]. Chronic and excessive dietary intake of these constituents may trigger a chronic inflammatory response in multiple tissue organs, which then develop into non-communicable diseases [4]. Both diabetes and dementia are examples of non-communicable diseases. It is well-known that the Western dietary pattern is strongly linked to the development of obesity and type 2 diabetes mellitus (T2DM) [5]. Other studies also report the association between adopting a Western dietary pattern and impairments in hippocampal-dependent learning and memory performance across the lifespan [6,7,8,9,10,11,12]. A clinical study demonstrates that a causative relationship by subjecting a group of healthy young adults (*n* = 102) to a short exposure (four days) to a high caloric diet. Healthy young adults are subjected to high saturated fat and added sugar breakfasts (53% total fats, 37.5% carbohydrates, 11.5% proteins) or a control diet (15.9% total fat, 31.8% carbohydrates, 51.3% proteins) for four days [13]. Four days after, individuals consuming a high caloric diet have lower retention scores in the Hopkins-Verbal Learning Test as compared to the controls, implicating a decline in hippocampal-dependent learning and memory performance. Of particular note, a negative correlation between retention score and blood glucose level is observed. Moreover, interoceptive sensitivity to satiety is also reduced in these individuals consuming high caloric diet, suggesting a lower appetitive control. Another clinical study further demonstrates a correlation between impaired appetitive control and decline in memory retention test score after one week of high caloric diet consumption (*n* = 110) [14]. The enigmatic relationship underlying high-caloric diet, obesity, diabetes, and hippocampal plasticity has been actively investigated in rodent studies.

Obesity could predispose individual to type 2 diabetes mellitus (T2DM), which is characterized by hyperglycemia and insulin resistance. Both peripheral [15,16] and central insulin resistance [17,18] are associated with cognitive impairment and neurodegenerative diseases. As supported by animal studies, progressive peripheral and central insulin resistance are linked to Alzheimer’s disease (AD) pathology and cognitive impairments [19,20,21,22,23,24]. Also, the AD-associated insulin signalling impairment in the brain resembles the dysfunction of insulin signalling in the peripheral organs, including the pancreas, adipose tissues, liver, and muscle, in diabetes [25,26,27,28,29]. β-amyloid accumulation is one of the pathologies found in the AD brains. In the form of amylin or islet amyloid polypeptide accumulation, amyloidosis is also observed in the pancreas under diabetic condition [30]. Notably, AD is classified as type 3 diabetes presenting impaired insulin signalling and neurological deficits in the brain [31]. It has been previously reported that adipokines secreted from the adipose tissue could influence brain plasticity and cognitive function in both physiological and pathological conditions, respectively [32,33]. Indeed, literature has reported that individuals with obese [34,35,36] and diabetes [37,38] could have a higher incidence of dementia, suggesting that early-life metabolic syndrome could contribute to dementia in the later life. 

Amble animal studies have shown that diabetic conditions are associated with insulin resistance and cognitive impairments [39,40,41,42]. Chronic consumption of high-fat diet impairs spatial learning and memory performance in the radial arm maze in juvenile mice [43]. Leptin receptor deficiency also impairs spatial memory in the Morris water maze task [44] and working memory in the Y-maze task, as well as disrupts prepulse inhibition in *db*/*db* mice [45]. The hippocampus plays an essential role in regulating spatial learning and memory processes, as well as affective behaviours [46]. Specifically, the dorsal hippocampus is involved in spatial learning and memory, while the ventral hippocampus is involved in mood regulation [47]. Conversely, impaired hippocampal plasticity by stress, ageing, or metabolic disorders could lead to learning and memory deficits, mood disorders, and eating disorders [48,49,50,51,52]. Diabetes-induced structural and synaptic deficits in the hippocampus are predisposing factors of learning and memory impairments and mood disorders [53,54,55,56,57,58,59,60,61]. The dentate gyrus (DG) of the hippocampus is one of the neurogenic zones with continuous generation of new neurons in the adult brains [62]. Animal studies have demonstrated that obese and diabetes suppress cell proliferation [58,63,64,65], neuronal differentiation [43,65,66], and cell survival [51] of the new-born cells in the hippocampal DG. Furthermore, obese and diabetic conditions also reduce dendritic complexity [44,67] and synaptogenesis [44,51] in the hippocampus, suggesting the hippocampus is highly vulnerable brain regions to metabolic dysregulation.

Long-term potentiation (LTP) and long-term depression (LTD) are cellular mechanisms underlying changes in synaptic plasticity [68] and learning and memory formation [69]. LTP impairment [44,70,71] and LTD facilitation [72] in the hippocampus are linked to learning and memory deficits in diabetic rodents [73,74,75]. Deficits in LTP aggravate with age [73], disease duration [76], and severity [77] of the diabetic condition. The role of glutamatergic transmission in LTP formation is critical, whereas reducing the extracellular level of glutamate at the synaptic cleft could impair LTP induction [78]. Glutamine-glutamate (Gln/Glu) ratio is an indicator of glutamatergic synaptic transmission with an implication of enhanced glutamatergic transmission in high Gln/Glu ratio [79]. Nevertheless, prolonged high-fat diet (HFD) for ten weeks reduces the Gln/Glu ratio in the hippocampus with substantial elevation in glutamate levels, but not glutamine levels [80]. Glutamate accumulation at the synaptic cleft may lead to excitotoxicity [81]; and thus, impairs LTP formation. Moreover, chronic high-caloric diet reduces protein expression levels of synaptic receptors, which could account for impairment in synaptic transmission in obese and diabetic conditions. To facilitate LTP, GluA1 subunit-containing AMPA receptors are rapidly recruited to the postsynaptic membrane upon stimulation by insulin or by NMDA receptor-mediated calcium influx [82]. Six-week HFD induces GluA1 palmitoylation and reduces phosphorylation, and consequently reduces AMPA receptor function [83], which could be linked to decreased postsynaptic contents of GluA1 and GluA2 in the hippocampus [84]. On the other hand, HFD promotes palmitoylation of NMDA receptor subunits (GluN2A and GluN2B) [83]. Specifically, eight weeks [85], but not six weeks [83] of HFD intake affects GluN2B expression, suggesting the time-dependent effect of HFD on NMDA receptor function. Of note, HFD also reduces the expression levels of PSD-95 (a postsynaptic scaffolding protein in glutamatergic synapses) and gephyrin (a postsynaptic scaffolding protein mediating aggregation of GABA_A_ receptors), suggesting that HFD could impair synaptic plasticity by modulating both glutamatergic and GABAergic function in the hippocampus [86]. 

Comorbid depression with diabetes and obesity is observed in humans [87,88]. Similarly, concomitant HFD feeding and corticosterone administration accelerate the onset of diabetes in a rat model [89], highlighting the neuroendocrine interaction of stress and high-caloric diet on promoting the pathogenesis of diabetes. Obese and diabetic animals show depression-like behaviors [90,91,92], which could be partly contributed by impaired hippocampal adult neurogenesis [93,94,95]. Both rodent models of depression [96,97,98,99] and diabetes [100,101,102] have shown a hyper-activated hypothalamic-pituitary-adrenal (HPA) axis, resulting in elevated glucocorticoid levels in response to stressors. The hippocampus exerts inhibitory feedback control to the activation of the HPA axis through GABAergic innervation to the paraventricular nucleus (PVN) of the hypothalamus [103,104,105]. Hippocampal dysfunction reduces the capacity to regulate the activity of the HPA axis. Four-week HFD increases serum levels of corticosterone and suppresses hippocampal newborn cell survival [64]. Exogenous administration of corticosterone suppresses hippocampal adult neurogenesis. It impairs dendritic complexity which is linked to increased depression-like behavior and impaired learning and memory [106,107,108], suggesting that high intake of dietary fat could dysregulate hippocampal neurogenesis through hyper-activating the HPA axis.

Recent studies have reported that an HFD feeding protocol as short as one week is sufficient to impair learning and memory performance, whereas hippocampal plasticity impairment and mood disturbances occur as the HFD feeding continues [109,110]. However, one-week HFD induces object recognition deficits, which can be restored by HFD withdrawal [111]. On the other hand, an eight-week dietary reversal from 16-week HFD can also restore learning and memory deficits and glucose intolerance, though insulin resistance is irreversible [112]. Uncontrolled food-seeking behaviors are observed in rodents consuming high-calorie palatable food [113,114] as well as in obese [115] and diabetic rodents [116]. The data collectively suggest that permissive obesogenic environment encourages hyperphagia-like behavior, which further promotes the progression of metabolic and neurocognitive diseases. Overproduction of corticosterone due to hyper-activated HPA axis promotes the secretion of the orexigenic neuropeptide called agouti-related peptide (ARP) in the hypothalamus, and thus promotes food consumption [117,118]. This finding could partly explain clinical observation in which individuals exposed to psychosocial stress and obesogenic environment have increased food consumption [119]. Furthermore, it is known that the hippocampus is involved in memory-related to eating episodes and food cue. For example, inhibiting GABA_A_ receptor in the dorsal hippocampus promotes postprandial sucrose consumption [120], whereas sucrose consumption promotes neuronal activation in the dorsal CA1 sub-region [121]. These findings have suggested that the dorsal hippocampus plays a role in regulating eating behavior [122]. It is also found that food consumption could be increased by manipulating the ventral hippocampus functioning through different approaches including lesion [123], optogenetic inactivation [124], local activation of leptin and GLP-1 receptors [125,126], inhibition of GABA_A_ receptor [127]. In sum, impaired synaptic plasticity in the hippocampus and hyperactivation of the HPA axis can lead to uncontrolled appetitive behaviors, which further aggravate the severity of metabolic syndromes [128,129].

The above studies have suggested the possible effects of obesogenic diet and diabetic condition on impairing neural plasticity in the hippocampus, which in turn could contribute to cognitive decline, emotional disturbances, as well as hyperphagia behaviors. Adipose tissue regulates hippocampal functions via releasing adipokines that can directly or indirectly modulate both neural plasticity [58,130,131,132,133,134,135,136,137,138] and neuroinflammation [44,55,139,140]. However, the potential mechanisms of how adipose tissue inflammation in obesity can progressively lead to neurodegeneration and neuroinflammation in the hippocampus have yet to be explored. In this review, we summarize how adipose tissue dysregulation and inflammation can lead to neurodegeneration and cognitive decline. Leptin resistance and inflammasome activation in the adipose tissue contribute to cognitive deficits, in which neural plasticity is impaired, and neuroinflammatory responses are activated in the hippocampus. Inflammation occurred in the adipose tissue suppresses the secretion of anti-inflammatory adipokines, such as adiponectin [141,142], that can directly affect hippocampal plasticity [132,143,144,145]. Though both impaired neuroplasticity and neuroinflammatory response are presented in diet-induced obese and diabetic models, microglial activation could play a predominant role in impairing hippocampal plasticity and cognitive function in these models. We also explored the possible role of microglia-mediated synaptic stripping in diabetic or obese. While withdrawing obesogenic diet could be one of the non-pharmacological interventions to improve metabolic profiles, physical exercise and balanced diets are known to be effective in ameliorating memory deficit and amyloid deposition in an AD mouse model [146]. Physical exercise is known as a non-pharmacological intervention to combat metabolic syndromes [147] and cognitive deficits [148,149] in obese and diabetic individuals. In response to physical exercise, multiple organs are known to secrete cytokines or metabolic hormones. The term “exerkine” has been used to describe secretory factors-induced by physical exercise acting on the brain and peripheral organs. Some of the exerkines are found to be pro-cognitive; therefore, this review also summarized the roles of some exerkines secreted from the liver, muscle, and adipose tissue on modulating neuronal metabolism, neuroinflammation and hence neuroplasticity that underlying changes in cognitive functions. 

## 2. Gut Is a Potential Origin of Chronic, Low-Grade Inflammation in Obesity and Diabetes

Chronic, low-grade inflammation in the adipose tissue is a characteristic of diet-induced obesity [150]. The critical effector that triggers diet-induced the adipose tissue inflammation is often masked. This is because metabolic and immunological complications have arisen from multiple organs in obesity, or when diabetes is diagnosed. Studies have suggested that the gut microbiome is inextricably linked to obesity. A pioneering animal study has reported that naïve recipients harbouring the gut microbiota from *ob*/*ob* mice have increased percentage body fat, increased energy consumption, as well as increased acetate and butyrate concentrations in the faecal samples [151]. This finding is echoed by a later study investigating the effect on adiposity by inoculating the microbiota from pairs of human twins, of whom one twin is obese, and the other is lean, in the germ-free mice. Upon the transplantation of microbiome, mice consume a low-fat (4%) and high-plant polysaccharides diet. However, mice become obese after human microbiome transplantation from the obese twin, whereas mice harbouring the human microbiome from the lean twin remains lean. When respective mice harbouring the lean and the obese microbiota are co-housed, both mice are resistant to obesity. The study further reports that the *Bacteroidetes* species in the gut microbiota from lean humans can resist the obese microbiota invasion [152]. These studies suggest that gut can be an origin of obesity.

High-fat diet induces endotoxemia with increased circulating lipopolysaccharides [153]. Lipopolysaccharide is the main component of gram-negative bacteria outer membrane, constituting a vast endotoxin reservoir in the gut. High-fat diet alters the ratio of gram-negative to gram-positive bacteria in the gut microbiome with an elevated composition of gram-negative bacteria [153,154]. Concomitantly, both dietary fat and intestinal dysbiosis reduce the integrity of the intestinal lumen [155,156]. The increased gut barrier permeability, also known as the leaky gut, results in the leakage of endotoxin [153,157]. The excess entry of gram-negative bacteria-derived lipopolysaccharides into the circulation results in endotoxemia and systemic inflammation [155,158,159,160].

Visceral adipose tissue is one of the target sites where lipopolysaccharides induce inflammation. 11-week lard diet increases the circulating lipopolysaccharide level and increases the expression levels of toll-like receptor 2 (TLR2) and toll-like receptor 4 (TLR4) when compared to mice receiving 11-week fish oil diet. Lipopolysaccharide binds to TLR4 on adipocyte, which in turn activates the Trif/MyD88/CCL2 signalling pathway [161,162,163]. TRIF and MyD88 are TLR adaptor molecules [164]. Chemokine CCL2 is a mediator of macrophage accumulation in white adipose tissue in obesity [165,166,167]. Interestingly, mice lacking TRIF and MyD88 are protected from lard diet-induced body weight gain and white adipose tissue inflammation. *Trif*-knockout and *MyD88*-knockout also prevent lard diet-induced CCL2 expression and inflammation in white adipose tissue as well as body weight gain. TLR4 recruits TRIF and MyD88, which promotes the expression of CCL2 in adipose tissue. The obesity-induced elevation of CCL2 level further recruits macrophage in white adipose tissue [168] (Figure 1). 

The condition of the gut microbiome is shown to affect hippocampal plasticity. Young microbiome-free recipients harbouring the gut microbiota from the old mice promotes hippocampal neurogenesis and longevity. Conversely, microbiome-depleted recipient transplanted with faecal microbiota of high-fat diet-fed donor presents greater anxiety-like behaviour, which is accompanied by the increased expressions of lymphocyte and microglial marker in plasma and whole brain [169]. Further evidence shows that faecal microbiota transplant from 24-month-old donor mice impairs spatial learning and memory performance and novel object recognition in the young recipient [170]. These findings suggest that gut dysbiosis induces cognitive impairment. It is also possible that the gut has direct communication with the hippocampus [171]. Circuit study reveals a more direct connection between the gut-hippocampus axis. The medial nucleus tractus solitarius (mNTS) receives gut vagal sensory input, whereas the mNTS connects the dorsal CA3 region of the hippocampus through the medial septum [172]. Both the elimination of gastrointestinal vagal afferents and efferents by subdiaphragmatic vagotomy and selective gastrointestinal vagal deafferentation by injecting saporin-conjugated cholecystokinin into the nodose ganglia impair spatial memory and contextual episodic memory. The impaired hippocampal-dependent learning memory performances are accompanied by reduced hippocampal BDNF and doublecortin levels [172].

## 3. From Inflammation in Adipose Tissue to Impaired Hippocampus Plasticity in Obese and Diabetic Conditions

Obesity induces chronic inflammation in adipose tissues due to infiltration and activation of macrophages [173]. During extreme obesity, it is estimated that these macrophages can take up over 50% of all cells in the adipose tissue [174] and are polarized to the pro-inflammatory phenotype upon activation [175]. Several fat transplantation studies have demonstrated that the inflammatory response in adipose tissue could be linked to obesity and diabetes-impaired hippocampal plasticity and cognitive functions [55,176]. Leptin, a pro-inflammatory cytokine secreted from the adipose tissue, is elevated in obese and diabetic conditions [177,178,179]. Transplantation of epididymal fat from *db*/*db* (leptin receptor deficiency) donor to wild-type naïve mice activates microglia and macrophages in the hippocampus, and reduces dendritic spine density in the granule neurons, as well as impairs hippocampal long-term potentiation (LTP) [55]. These structural changes could be linked to spatial memory deficits in the hippocampal-dependent tasks [55]. Conversely, epididymal-lipectomy in *db*/*db* donor rescues spatial memory deficits, suppresses neuroinflammatory response, and restores hippocampal plasticity [55]. These findings have suggested that leptin resistance in adipose tissue could be a contributing factor to hippocampal plasticity deficits, and hence cognitive impairment.

The high-fat diet promotes the generation of reactive oxygen species (ROS) in the adipose tissue [180]. ROS can trigger nod-like receptor family, pyrin domain-containing 3 (NLRP3)-containing inflammasome to activate caspase-1 and secrete interleukin 1β (IL-1β) [181]. HFD induces obesity in *Nlrp3* global knockout mice but prevents the development of adipose tissue inflammation and insulin resistance [182,183]. Increased circulating IL-1β level is associated with cognitive impairment in diabetes [184]. Visceral adipose tissue transplantation experiment in mice suggests that adipose tissue inflammation has a direct linkage to cognitive impairment with the associated neuroinflammatory response. Transplanting visceral adipose tissue from mice fed with 12-week HFS to wild-type recipient impairs spatial memory, activates microglial and increases hippocampal IL-1β levels [176]. On the contrary, fat transplantations from HFD-fed *Nlrp3*^−/−^ donors do not trigger neuroinflammatory response and memory deficits in recipients [176]. Other studies have shown that NLRP3-inflammasome/IL-1β signaling contributes to synaptic deficits in the hippocampus. The electrophysiological recording reveals that NLRP3 impairs LTP formation through IL-1 receptors [176]. Conversely, neutralizing IL-1 receptor rescues deficits in structural and synaptic plasticity in the hippocampus of the *db*/*db* mice [55]. In sum, these results have suggested that activation of NLRP3-inflammasome/IL-1β signalling could activate neuroinflammatory response and impair synaptic plasticity, leading to cognitive impairment in diet-induced obesity or diabetic condition (Figure 1).

Diet-induced obesity up-regulates the secretion of pro-inflammatory cytokines, whereas inflammatory response in adipose tissue further suppresses the secretion of adipocyte-derived anti-inflammatory cytokines, such as adiponectin [185]. Adiponectin is the most abundant adipokine in the bloodstream, which is secreted by mature adipocytes [186]. Adiponectin is an insulin-sensitizer by promoting glucose and fatty acid metabolism upon feeding [187]. In obese individuals, adiponectin secretion from white visceral adipose tissue decreases as adiposity increases [187]. Reduced insulin sensitivity, together with chronic inflammation, could progressively lead to systemic and central insulin resistance [24,188] (Figure 1). Adiponectin can cross the blood-brain barrier [130,189], suggesting its potential role in promoting insulin sensitivity in the brain. The functional role of adiponectin in the brain has recently been found using adiponectin knockout mouse models. Adiponectin activates AMPK to increase insulin sensitization in hippocampal and cortical neurons via suppressing Akt/GSK3β signaling [144]. Conversely, adiponectin deficiency disinhibits GSK3β-mediated cleavage of amyloid precursor protein and promotes plaque deposition [144], which in turn leads to neuroinflammation and hippocampal-dependent learning and memory deficits [144]. Adiponectin also mediates neuroinflammatory response through AdipoR1/NF-κB pathway [190]. Adiponectin deficiency induces microgliosis in the hippocampus and hypothalamus in an association with increased pro-inflammatory cytokine secretion [190]. Adiponectin also promotes hippocampal synaptic [132] and structural plasticity [131], which are abolished in adiponectin-deficient mice. Both obese and diabetic mice show reductions in adiponectin levels in the hippocampus [58,191]. These studies have collectively suggested that obesity and diabetes alter adiponectin secretion, which could consequently result in neurodegeneration and cognitive deficits.

Metabolic syndromes perturb adipose secretion of pro- and anti-inflammatory cytokines as well as adipose tissue inflammation. These perturbations not only affect the peripheral metabolism, but also trigger microglial activation in the hippocampus, and hence impair hippocampal plasticity. Studies have shown that withdrawal from the high-fat diet is an effective non-pharmacological intervention for improving learning and memory deficits [111,112]. Another study has also demonstrated that a feeding protocol with a 12-week high-fat diet, followed by an 8-week low-fat diet reduces microglial activation and increases spine density in the hippocampus [192]. The beneficial effect of dietary reversal is accompanied by reduced weight gain and fat masses [192]. Little is known about the interplay of microglial activation and adult neurogenesis in the obese and diabetic brain. However, it is shown that aberrant microglial activity may predominate the interplay by internalizing synaptic terminals upon chronic HFD while switching from HFD to a low-fat diet (LFD) attenuates synaptic internalization [192]. A recent study has reported that fractalkine receptor (CX3CR1) is involved in the microglial-mediated synaptic stripping in obese mice [193]. Obesity induces the expression of the phagocytic marker in hippocampal microglia [193], whereas *Cx3cr1*-haploinsufficiency counteracts spatial learning and memory deficits, microglial activation, and dendritic spine loss in the hippocampus [193]. CX3CR1 mediates microglial motility and activation [194]. Pharmacological blockade of microglial phagocytosis by annexin-V or suppressing microglial activity by minocycline prevents obesity-induced spine lost and cognitive impairment [193]. In sum, these findings have suggested the predominant role of microglia in inducing deficits in hippocampus structural plasticity in obesity (Figure 2).

## 4. Neuronal Mitochondria Are Involved in Obesity- or Diabetes-Induced Neuroinflammation and Hippocampal Impairment

Restoring or enhancing brain energetics has been proposed to be a potential therapeutic approach to halt neurodegenerative diseases of ageing [195]. Mitochondrial perturbation is observed in obese and diabetic conditions [196,197,198]. Independent of genetic factor, hyperglycemia contributes to the generation of reactive oxygen species (ROS) [199,200]. Excessive ROS production can induce mitochondrial DNA damage, lipid peroxidation, as well as oxidative phosphorylation (OXPHOS) [201]. These perturbations could increase the production of ROS and reduce oxidative capacity [202]. 

Diabetes-induced learning and memory deficits are associated with reduced mitochondrial density and ATP formation in neurons [203]. PGC-1α is a master regulator of mitochondrial biogenesis [204]. HFD induces insulin resistance and remarkably reduces expressions of transcriptional co-regulator of mitochondrial biogenesis (PGC-1α), mitochondrial transcription factor A (TFAM) and the mitochondrial OXPHOS complexes in the hippocampus [205]. Prominently, PGC-1α mediates the formation and maintenance of neuronal dendritic spines. siRNA-mediated PGC-1α silencing reduces dendritic spine density of granule neurons in the mouse hippocampal DG region. Consistently, PGC-1α silencing not only reduces dendritic spin density in primary hippocampal neurons [206], but also reduces mitochondrial density and ATP formation [206]. Conversely, activating BDNF/PGC-1α signaling cascade promotes synaptogenesis in hippocampal neurons [206], suggesting the role neuronal mitochondria on regulating synaptic plasticity through PGC-1α signaling.

Other studies have further revealed the alternative linkage between mitochondria-mediated dendritic spine integrity and microglial activity. Photo-ablation of mitochondria triggers dendritic spine clearance and dendritic retraction in primary hippocampal neurons, which activates caspase-3 in hippocampal dendrites [207]. Increased caspase-3 expression in the hippocampus is observed in diabetic rodent models [208,209]. Phospholipid scramblase and flippase are two families of enzymes that catalyze the translocations of phospholipids between the two monolayers of the cell membrane. Caspase-3 can, in turn, activate phospholipid scramblase and inhibit flippase [210,211], which exposes phosphatidylserine on the synaptic membrane. Microglia can recognize phosphatidylserine as an ‘eat-me’ signal and engulfing degenerating synapse [212]. Caspase-3 activation in the hippocampus of diabetic rodents may trigger a similar mechanism to recruit microglia for phagocytosis of damaged dendritic spines. Furthermore, autophagy could ameliorate the diabetes-induced metabolic crisis, apoptosis, and necrosis in neurons [213] (Figure 2). 

In addition, insulin resistance could be linked to mitochondrial malfunction [214]. Adiponectin increases insulin sensitivity. Muscle-specific knockout of adiponectin receptor 1 (*AdipoR1*) results in systemic insulin resistance and hyperglycemia [215], resembling diabetic condition. Reduction in mitochondrial content and activity in the skeletal muscle could be linked to decreased exercise endurance in *AdipoR1* knockout mice [215]. Mitochondrial activity and oxidative phosphorylation in skeletal muscles are found to be regulated by adiponectin/AdipoR1/AMPK/SIRT1/PGC-1α signalling pathway [215]. A proteomic study has revealed that *AdipoR1* knockout impairs signalling pathways that are important for mitochondrial functioning, including as oxidative phosphorylation, TCA cycle, β-oxidation [216]. The expression of nuclear respiratory factor 1 (NRF1), a downstream target of PGC-1α and involves in mitochondrial biogenesis, OXPHOS, and ROS scavengers [217], is downregulated in both *AdipoR1*^−/−^ and HFD-fed mice, [216]. Adiponectin deficiency results in central insulin resistance [144], impairment in hippocampal plasticity [131,132], and cognitive function [132,144]. Adiponectin deficiency triggers Aβ oligomerization [144], whereas Aβ assault reduces mitochondrial membrane potential in primary hippocampal neurons which could be restored by activation of adiponectin signalling using receptor agonist, AdipoRon [218]. *AdipoR1* knockout mice also display cognitive impairment [219]. These findings have provided potential linkage of how adiponectin signalling regulates mitochondrial function. The sophisticated mechanisms of how adipokines modulate neuronal mitochondria health and consequently affects neuronal functions warrant further investigation. 

## 5. Excessive Palmitate Consumption from Diet Triggers a Direct Neuroinflammatory Response in the Hippocampus

Palmitate is the most abundant saturated fatty acid present in the circulation [220] and cerebrospinal fluid [221]. Increased brain uptake and accumulation of palmitate is reported in individuals with obesity and metabolic syndromes [222]. Moreover, palmitate is increased in the cerebrospinal fluid of overweight and obese humans [223]. Studies have suggested that palmitate impairs synaptic plasticity by elevated microglial activity. Palmitates suppress LTP formation in the Schaffer-collateral path [223]. Local infusion of palmitate in the cerebral ventricles impairs learning and memory performance in the object recognition task, object location task, step-down task, and the Barnes maze task [223]. In vitro study suggests that IRS-1 signalling is suppressed in the hippocampal neurons by the microglial-derived TNF-α [223], implicating palmitate-induced microglial activities may reduce insulin sensitivity in the hippocampal neurons. In another study, exosome fraction, which is isolated from palmitate-stimulated microglia in vitro, induces an immature dendritic spine phenotype in primary hippocampal neurons [224]. Lastly, high-fat diet also induces similar predominance of immature dendritic spines from CA1 neurons alongside with diminished levels of the scaffold protein Shank2 and impaired spatial memory performance [224]. These studies highlight the microglia-neuronal communication through exosomes, where palmitate-induced microglial inflammation may adversely influence on spine growth in the neighbouring neurons. 

## 6. Obese and Diabetic Conditions Inhibit Hippocampal Feedback Control of the HPA Axis

An epidemiological study has reported that type 2 diabetic individuals have a higher prevalence of hypercortisolemia [225] while hypercorticosteronemia is also observed in diet-induced obese and genetically diabetic rodent models [226,227]. The increased activity in the hypothalamic-pituitary-adrenal axis (HPA) is a major cause of elevated cortisol or corticosterone secretion from the adrenal gland [228]. Lesion studies have illustrated that the hippocampus exerts efferent inhibitory control of the HPA axis. It is reported that total hippocampectomy elevates the expression of corticotropin-releasing hormone in the hypothalamic paraventricular nucleus (PVN) [229], whereas glucocorticoid secretion is markedly increased in rats bearing lesion in the ventral hippocampus [104]. Besides, recent circuit studies have revealed that the glutamatergic projections from ventral hippocampus synapses on the bed nucleus of striatum terminalis (BNST), whereby the BNST GABAergic projection elicits an inhibitory control over PVN [124,230]. It is reasonable to speculate that obese- or diabetes-induced hippocampal impairment may abolish the efferent inhibitory control over the HPA axis leading to hypercorticism. The presence of glucocorticoid receptors [231] and the ability that corticosterone can cross the blood-brain barrier [232] further implicate that the inhibitory action of hippocampus on HPA axis involves feedback inhibition. The feedback inhibition mechanism is supported by the fact that deletion of glucocorticoid receptors in the corticolimbic forebrain, including the hippocampal region, prolongs HPA axis activation [233,234].

Acute exposure of the high-fat diet for three days is sufficient to elevate corticosterone levels in rats and triggers neuroinflammatory responses with increased inflammasome-associated NLRP3 and CX3CR1 expressions in the hippocampus [102]. Acute HFD exposure leads to a higher vulnerability to a lipopolysaccharide-triggered inflammatory response with much higher hippocampal IL-1β and IL-6 expressions [102]. Independent of hypercorticosteronemia, systemic administration of a blood-brain barrier permeable glucocorticoid receptor antagonist: mifepristone [235] (also known as RU-486; 50 mg/kg s.c., two doses in three days) suppresses the expressions of neuroinflammatory markers in the hippocampus and attenuates lipopolysaccharide-induced neuroinflammation [102]. In addition to the pro-inflammatory effect of corticosterone, corticosterone impairs hippocampal plasticity, abrogates learning and memory performance, and reduces BDNF expression [106,108,236,237,238]. Conversely, mifepristone administration not only rescues memory deficits in an object recognition task but also restores LTP deficit in the Schaffer-collateral path in three-week-old juvenile rats exposed to the high-fat diet for 7 to 9 days [239]. Potentially, the high-fat diet-induced corticosterone surge impairs hippocampal function and plasticity, which in turn weakens the efferent inhibitory control over the HPA axis, where hypercorticosteronemia is observed in obesity and diabetes.

Conversely, hypercorticosteronemia-induced neuroinflammation and impaired hippocampal plasticity in obesity and diabetes can be intervened by normalizing the circulating corticosterone level or blocking the central action of corticosterone. Systemic administration of metyrapone reduces circulating corticosterone level by suppressing corticosterone synthesis [240,241]. In dependent of insulin sensitivity in the hippocampus, metyrapone administration in five-week-old *db*/*db* mice (100 mg/kg i.p., three weeks) reduces microgliosis and suppresses the expressions of IL-1β and TNFα in the hippocampus [242]. In vitro study has further illustrated that metyrapone administration reduces the pro-inflammatory M1 microglia population in the hippocampus of *db*/*db* mice as well as reduces microglial sensitivity against lipopolysaccharide assault in primary microglia culture [242]. These findings emphasize that the direct pro-inflammatory effect of corticosterone in the hippocampus regardless of a short-term high-fat diet exposure context or a genetically diabetic background. In *db*/*db* and streptozotocin-induced diabetic mice, adenectomy with concomitant corticosterone replacement restores cell proliferation and LTP formation in the hippocampus as well as improves learning and memory performance [44]. These restorations are linked to the reduction in circulating corticosterone levels and are independent of diabetes-induced hyperglycaemia [44]. In situ hybridization further reveals that adenectomy and corticosterone replacement restores BDNF-TrkB expressions in the hippocampal dentate region of *db*/*db* mice [243]. Moreover, suppression of corticosterone synthesis by systemic metyrapone administration in *db*/*db* mice rescues spatial memory deficits in concurrent with restored LTP formation and spine density of the granule cells in the hippocampus [227]. Systemic metyrapone administration fails to rescue the corticosterone-induced memory impairment, LTP deficits, and spine loss when corticosterone is locally infused in the hippocampus [227], suggesting the direct central action of corticosterone.

Corticosterone fails to induce spatial memory and synaptic impairments in *db*/*db* mice when glucocorticoid receptors are knocked down in the hippocampus [227], suggesting the potential role of glucocorticoid receptor signalling in the corticosterone action. Corticosterone impairs the hippocampal BDNF signalling pathway in *db*/*db* mice by two means. Firstly, the activation of glucocorticoid receptor transcriptionally suppresses the hippocampal BDNF expression. Concomitantly, the expressions of BDNF receptors are altered in *db*/*db* mice. While BDNF has a higher affinity for TrkB receptor, which promotes neural plasticity [244], it has a lower affinity for the P75 neurotrophin receptor (P75^NTR^) [245], which constrains plasticity [246]. TrkB expression is down-regulated, and P75^NTR^ expression is up-regulated in the hippocampus of *db*/*db* mice, while pharmacological inhibition of corticosterone synthesis by metyrapone can re-activate the BDNF/TrkB cascade and bypass the P75^NTR^ pathway. These studies demonstrate the complexity of the negative feedback mechanism of the hippocampus-HPA axis in response to obesity and diabetes-induced hypercorticosteronemia, in which the action of corticosterone on activating inflammasome-associated response and suppressing BDNF signaling induces neuroinflammation and plasticity deficits in the hippocampus (Figure 3). 

## 7. High-Fats and Low-Carb Ketogenic Diet Elicits Neuroprotective Effects by Promoting Mitochondrial Dynamics and Reducing Oxidative Stress

There are misconceptions that high fat and dietary carbohydrate intake is directly linked to metabolic syndrome. Instead, high-fat diet elicits obesogenic effect when the caloric balance is upset by a high caloric intake versus a low energy expenditure, for example, physical inactivity. Counterintuitively, a high-fat, low-carbohydrate ketogenic diet (up to 90% fats in rodent studies) has been employed as a non-pharmacological diet intervention for controlling weight gain, glycemic index, and lipid content [247] and prevention for the recurrences of epilepsy [248]. 

Adopting a high-fat and very low-carbohydrates diet mimics the metabolic states of fasting and prolonged physical exercise, and elevates the levels of ketone bodies, known as physiologic ketosis [249]. When glucose reserve is depleted, lipids become the main source for ATP biosynthesis. Triacylglyceride is catalyzed into acetyl-CoA through β-oxidation, which then metabolized into ketone bodies, including acetone and β-hydroxybutyrate. β-hydroxybutyrate is the substrate for ATP biosynthesis. Ketosis in physiological state is different from diabetic ketoacidosis. Diabetic ketoacidosis, on the other hand, is a life-threatening complication of diabetes mainly due to the massive breakdown of lipids in adipose tissue induced by insulin deficit [250]. The high free fatty acid in the circulation accounts for approximately 80% of energy through β-oxidation, leading to the accumulation of ketone bodies (20–25 mM) at a level much higher than that during fasting or consuming a ketogenic diet (4–6 mM) [251,252,253,254,255,256]. 

Ketogenic diet elicits anti-inflammatory effect on adipose tissues under short-term consumption. Short-term exposure to a ketogenic diet (89.5% fats; 0.1% carbohydrates) for one week promotes ketogenesis with increased circulating β-hydroxybutyrate level in fed and fasting states [257]. Short term ketogenic exposure reduces the population of pro-inflammatory macrophages in the adipose tissue and down-regulates the *Nlrp3* and *Il1b* expressions in the adipose-resident immune cells. In another study, it is also reported that four-week ketogenic diet (93.4% fat, 1.8% carbohydrate) down-regulates the expressions of inflammatory markers, such as *Il6*, *Tnf*, and *Nlrp3*, in the epidydimal white adipose tissue [258]. On the contrary, chronic exposure to a ketogenic diet (89.5% fats; 0.1% carbohydrates) for four months impose a detrimental effect on adipose tissue metabolism [257]. Chronic exposure promotes body weight gain and hyperglycemia [257]. Macrophage population is significantly increased in association with up-regulated inflammatory marker expressions, including *Mcp1*, *Tnfa*, and *Il1b*, in the epididymal fat tissues [257]. Still, the circulating β-hydroxybutyrate level remains high after four months of consuming a ketogenic diet. After that, an independent study attempts to harness the beneficial effect of ketogenesis by adopting a cyclic ketogenic diet paradigm. From 12-month-old onwards, naïve mice receive an alternating ketogenic diet (90% fats; 0% carbohydrates) and standard chow (13% fats; 77% carbohydrates) weekly until 36 months old [259]. Cyclic ketogenic diet elevates blood β-hydroxybutyrate levels but does not induce body weight gain. Aged mice have better memory recall in place avoidance test (at 22 to 24 months old) and object recognition task (at 28 to 30 months old) on a cyclic ketogenic diet than receiving standard chow. Also, the cyclic ketogenic diet reduces early mortality in naïve mice at 12 to 24 months old.

The dynamics of the NAD^+^/NADH ratio implies mitochondrial activity with increased mitochondrial fission increases the NAD^+^ levels. The two-week ketogenic diet (6:1 fats-to-carbohydrates and proteins ratio) elevates hippocampal NAD^+^/NADH ratio in 11 to 14-week-old rats [260]. In vitro study suggests that β-hydroxybutyrate may contribute to the increase in NAD^+^/NADH ratio [261]. Poly-ADP ribose polymerase-1 (PARP1) serves as a DNA damage sensor and participates in DNA repair [262]. While NAD^+^ is a substrate for PARP1, DNA damage triggers PARP1-induced NAD^+^ depletion [263]. The ketogenic diet, on the other hand, reduces hippocampal PARP-1 and 8-hydroxy-2′-deoxyguanosine levels, suggesting a potential reduction in DNA damage or enhanced DNA repair activity. SIRT-1 activity reduces cell death and inflammation [264,265] and promotes neuronal survival [266,267,268], while ketogenic diet promotes the activities of nuclear sirtuins as well as the hippocampal *Sirt1* expression [260]. Altogether, the findings have suggested that the ketogenic diet could exert neuroprotective by promoting mitochondrial dynamics and reducing oxidative stress in neurons, which may serve as a dietary intervention to prevent cognitive decline. 

Regardless, the effect of a ketogenic diet (84% fats; 0% carbohydrates) presents an adverse effect in the transgenic mice expressing a mutated mitochondrial DNA repair enzyme (mutUNG1) selectively in forebrain neurons [269]. The continuous ketogenic diet for two to four months reduces hippocampal size in mutUNG1 mice. Immunofluorescence shows a reduction in neuronal marker expression and an elevation of astroglia marker expression in the hippocampus upon a ketogenic diet in mutUNG mice. The increased SOD2/VDAC1 ratio as well the increased SIRT1 and FIS1 expressions in the hippocampus suggests ketogenic diet activates mitochondrial antioxidant defences and mitochondrial fission in mutUNG mice. Alterations in mitochondrial morphology, as well as the distribution of mitochondria within a neuron, are observed. For example, ketogenic diet induces swelling of mitochondria in mutUNG mice, whereas mitochondria are accumulated in the soma with reduced mitochondrial density in the presynaptic terminal of the pyramidal neurons in the hippocampal CA1. Lastly, electronic microscopy illustrates that ketogenic diet reduces the density of GluN2A and GluN2B subunits while increases the density of GABA_A_ α_1_ subunits in the hippocampal CA1 of mutUNG1 mice. This study calls for rectification on whether a ketogenic diet is a proper way to augment neurodegeneration and cognitive impairments in association with severe mitochondrial dysfunction. 

It is noteworthy that the peripheral metabolic, neuroprotective, and pro-cognitive effects of the ketogenic diet have called for substantial investigations. The ketogenic response is effective under low blood glucose and low insulin levels. Hyperglycemia and insulin resistance are hallmarks of diabetic patients, and these conditions are often shared in AD patients and older adults.

## 8. SIRT1/SIRT3 Axis Potentiates the Pro-Cognitive Effects of Caloric Restriction and Intermittent Fasting

Sirtuins (SIRT) are NAD^+^-dependent deacetylases [270]. SIRT1 is a cytosolic and nuclear sirtuin modulates transcription factors through histone deacetylation [264,271,272,273,274]. SIRT3 is a mitochondrial sirtuin regulating β-oxidation [275]. SIRT1/SIRT3 axis aims to promote energy storage intracellularly and to resist oxidative stress [276,277].

In vivo studies illustrate that SIRT1 is a mediator of caloric restriction-induced mitochondrial biogenesis in mice [278,279,280], while *Sirt1* knockout abrogates the metabolic benefits of caloric restriction [273,281]. In the hippocampus, 23 weeks of HFD reduces the expressions of *Ppargc1a*, *Ppp1cb*, *Reln* and *Sirt1* [282]. Increased DNA methylation and decreased DNA hydroxymethylation are observed in the promoter region of *Sirt1*, implicating a diet-induced epigenetic modulation of *Sirt1* [282]. On the contrary, caloric restriction promotes learning and memory performance as well as hippocampal synaptic plasticity by upregulating signaling transducers involved in mitochondrial biogenesis [283], including CREB phosphorylation and increased expressions of SIRT1, PGC-1α, nitric oxide synthase, and phosphoenolpyruvate carboxykinase in the hippocampus [283]. The activation of a mitochondrial biogenesis signaling pathway is associated with better learning and memory performance in the novel object recognition task and enhanced LTP formation in the Schaffer-collateral path in a neuronal-specific CREB-dependent manner [283]. 

SIRT3 is one of the SIRT1 deacetylation targets. HFD-induced obesity and ageing obliterate SIRT1-mediated deacetylation of SIRT3, which reduces SIRT3 activity and stability in the mouse model [284]. Intermittent caloric restriction, as short as one week to one month, promotes the hippocampal SIRT3 expression [285]. Besides, the anxiolytic and pro-cognitive effects of intermittent caloric restriction are abolished when *Sirt3* is selectively knockout in the hippocampal neurons. Intermittent caloric restriction further impairs LTP formation in the Schaffer-collateral path of the *Sirt3*-deficient mice. Significantly, intermittent fasting reduces seizure incidence, restores hippocampal-dependent spatial memory performance, as well as rescues LTP deficits in *App^NL-G-^*^F^ mice of AD.

SIRT3 can deacetylate LKB1, resulting in AMPK activation and mitochondrial biogenesis [286]. Four-week intermittent fasting activates AMPK/PGC-1a/mTORc/COX4/ND-1 signaling cascade, suggesting that intermittent fasting promotes mitochondrial biogenesis in the hippocampus. The increased mitochondrial DNA content further evidences this result in the hippocampus in *db*/*db* mice [287]. Essentially, intermittent fasting also exerts pro-cognitive and neurotrophic effects by restoring diabetes-induced hippocampal-dependent spatial memory deficits and the morphology of postsynaptic density of the neurons in the hippocampal CA1 region in addition to the activation of BDNF/pCREB/PSD-95 signaling cascade [287].

In sum, SIRT1/SIRT3 axis is one of the molecular targets of caloric restriction-induced pro-cognitive and neurotrophic effects, partly due to its functional role in regulating mitochondrial biogenesis.

## 9. Physical Exercise-Induced Mediators on Promoting Metabolic and Synaptic Function in the Brain 

Adipose tissue secrets IL-1β that can trigger the neuroinflammatory response in obesity- and diabetes, it also secrets endocrine hormones such as leptin [137] and adiponectin [58,130] to mediate pro-cognitive effects of physical exercise. The decrease in adiponectin levels in diabetic and obese patients could be due to chronic inflammation of adipocytes [288,289]. Accumulating evidence has suggested the function role of adipose tissues on influencing cognitive function, metabolic function, and neural plasticity. Physical exercise can effectively attenuate metabolic syndromes in obesity and diabetes by improving lipid profile, adipose tissue inflammation, and insulin resistance [290,291,292]. Wheel running is also shown to ameliorate memory deficit and amyloid deposition in HFD-fed AD mouse model [293]. Notably, physical exercise is not only useful in combating a wide range of neurocognitive and neuropsychiatric symptoms in obese and diabetic rodents, including improving learning and memory deficits [294,295,296,297,298,299,300,301,302,303], reducing depression-like behaviours [90,304], and improving uncontrolled appetitive behaviours in hyperphagia suppression test [305,306,307]. Restored adult neurogenesis, improved synaptic plasticity, and reduced neuroinflammation may altogether contribute to cognitive improvements in obese and diabetic rodents [66,67,295,308,309,310,311,312]. Mitochondria perturbation could disrupt metabolism from cellular to system levels [313]. The physical exercise shows opposite effects to promote cognitive function by promoting mitochondrial functions and reduce neuronal apoptosis in the hippocampus of mice fed with HFD [299]. 

Emerging data have found that metabolic factors secreted by peripheral organs are not only involved in regulating metabolism but also have a direct effect on neurodegeneration in association with metabolic disorders. Exerkines, including apelin secreted from the adipose tissues, β-hydroxybutyrate secreted by the liver, Fibronectin type III domain-containing protein 5 (FNDC5) secreted from the muscles, are involved in bridging the communication between the peripheral organs and the hippocampus. Functions of these factors on modulating hippocampal plasticity, neuroinflammation, and metabolic function in the brain are discussed below (Figure 4).

### 9.1. Apelin

Apelin is initially known as an adipokine [314] which is secreted by adipocytes. Hyperinsulinemia induces high circulating apelin levels in obesity [314]. Later studies have suggested apelin as a myokine because muscle contraction in young mice induces apelin secretion from the muscles [315]. Ageing suppresses apelin secretion from skeletal muscle while delivering exogenous apelin can rejuvenate the skeletal muscle strength in aged mice and apelin-deficient mice by promoting mitochondrial activity [315]. Overexpression of apelin also prevents HFD-induced obesity by promoting mitochondrial biogenesis and increasing energy expenditure in skeletal muscle [316]. Brain SIRT1, a mediator of neuroprotection and metabolic homeostasis [317], can be activated by apelin [318] and physical exercise [319,320]. Apelin receptor is abundantly expressed in rodent brains, including the hippocampus, hypothalamus, substantia nigra, entorhinal cortex [321]. Apelin suppresses inflammation-related NF-κB signaling by activating SIRT1 in the hippocampus [318]. Furthermore, apelin modulates microglia activation by attenuating the polarization of microglia to M1 pro-inflammatory phenotype under chronic stress condition [322]. Other than suppressing inflammatory responses, treatment with recombinant apelin-13 elicits anxiolytic [318] and antidepressant effects [322], and reduces the inflammatory response. Genetic knockout of *Nlrp3*, downstream target of NF-κB, obliterate chronic stress-induced inflammatory response and depressive behaviors [323]. Activation of SIRT1 is required to promote mitochondrial biogenesis as well as lipid and glucose metabolism [324,325]. In vitro study reveals mitochondria could mediate the neuroprotective effect of apelin-13. Apelin-13 preserves mitochondrial integrity and reduces the production of reactive oxygen species [ROS], neuronal apoptosis and NMDA-induced excitotoxicity in primary cortical neurons [326]. However, it is still largely unknown whether the antidepressant and anxiolytic effects of apelin are mediated by modulating microglia activity in the brain.

### 9.2. Irisin 

Irisin or fibronectin type III domain-containing protein 5 (FNDC5) is a myokine secreted by muscle cells. It can stimulate adipocyte browning and thermogenesis [327,328]. Diabetic patients show significant reductions in circulating levels of irisin and expression levels of *Fndc5 in* adipose tissue. Conversely, the expression level of *Fndc5* in skeletal muscle is upregulated in diabetic conditions. In obese mice, overexpression of *Fndc5* or subcutaneous administration with recombinant irisin reduces hyperglycemia, hyperlipidemia, and improves insulin resistance [329]. Voluntary wheel running promotes *Fndc5*/*Pgc1α*/*Erra* expressions in both the quadriceps and hippocampus and upregulates *Zif268*, *cFos*, and *Arc* expressions in the hippocampus, implicating an exercise-induced neuronal activation [330]. Intrahippocampal infusion of irisin into the dentate region promotes spatial and avoidance learning in naïve Wistar rats [331], possibly by promoting LTP formation [332]. Congruently, over-expression of irisin in the brain can restore hippocampal BDNF levels in concurrent with HFD/streptozotocin-induced diabetic condition in SD rats [333]. Irisin can cross the BBB [330]. However, peripheral neutralization of FNDC5 abolishes the pro-cognitive and neurotrophic effects of swimming exercise [334], suggesting peripheral irisin could mediate muscle-brain crosstalk. BDNF is a downstream target of irisin, while glucose metabolism is regulated by irisin in the brain [333]. Peripheral overexpression of *Fndc5* also promotes hippocampal BDNF levels [330]. It is noted that FNDC5 can also be synthesized in the brain. A recent study has suggested that exercise-induced lactate secretion from the muscle or exogenous lactate delivery promotes learning and memory through activating hippocampal endothelial monocarboxylate transporters/SIRT1/PGC-1α/FNDC5/BDNF signaling pathway [335]. Another study investigating the role of irisin on mood regulation demonstrates that irisin is a mediator of exercise-induced antidepressant and neurotrophic effect, while the irisin mechanism of antidepressant action requires more in-depth investigations [336]. These studies have highlighted the pro-cognitive [335,336] and neurotrophic effect [336,337] of myokines on the hippocampus. 

### 9.3. Lactate

Lactate is another myokine that is released from muscle in response to physical exercise. Lactate can cross the blood-brain barrier via endothelial monocarboxylate transporters [338,339]. The lactate action could be elicited by activating the lactate receptor HCAR1 in the brain. HCAR1 is present in the fibroblasts and ependymal cells in the brain [340]. HCAR1 mediates the lactate-induced neuronal excitability in primary cortical neurons [341] as well as high-intensity exercise-induced or lactate-induced angiogenesis in the brain [342]. Interestingly, systemic lactate administration partially mimics the wheel running effect on modulating the gene expression that regulates mitochondrial biogenesis in the liver and the whole brain. Lactate administration mimics the effect of exercise by upregulating the gene expressions of PGC-1α but downregulating the expressions of PGC-1β and NRF-1 in the liver. Besides, both lactate administration and exercise increase the PGC-1 related co-activator (PRC)/vascular endothelial growth factor A (VEGF-A) expression in the brain. Of note, exercise increases the mitochondrial DNA copy number in the brain and reduces TNFα expression but not after lactate administration [339]. Other pro-cognitive effects, including anti-depressant effects, are illustrated upon single-dose of lactate administration in corticosterone-induced stressed mice as well as after chronic lactate administration in naïve mice [343].

An earlier study has demonstrated the involvement of lactate in promoting neurogenesis and vascularization in the postnatal mouse brain [344]. Extracellular electrophysiological recordings have further revealed that lactate promotes synaptic plasticity at the Schaffer-collateral path [345]. In the CA3 pyramidal neurons, lactate acts on the postsynaptic lactate receptor and may facilitate AMPAR insertion in a PI3K-dependent manner, which is evidenced by increased intrinsic excitability of CA3 pyramidal neurons [345]. Potentially, lactate promotes NMDAR-mediated calcium ion influx through PI3K/PKC phosphorylation, as illustrated by an increased EPSP-spike coupling [345]. 

Lactate was known as a by-product of anaerobic respiration, but it is also a source of energy, a gluconeogenic precursor, and a signaling molecule [346]. Since then, the concept of lactate shuttle has been raised with two highlights. First, lactate can be exported and consumed by the neighboring cells through lactate receptors [paracrine route] or by a distal organ through circulation [endocrine route] [346]. Second, the role of lactates shifts from glycolytic to oxidative in the recipient tissues and organs [346]. The exercise-induced lactate/MCT/SIRT1/PGC-1α/FNDC5/BDNF signaling transduction has exemplified the endocrine perspective of lactate shuttle [335]. Also, several reviews have emphasized the lactate shuttle involving in synaptic plasticity and astroglial-neuronal metabolism [347,348,349,350,351]. Lactate enters the presynaptic neurons through monocarboxylate transport 2 (MCT2) [352]. In the presynaptic neurons, lactate plays an oxidative role in fueling synaptic plasticity [353]. During synaptic activation, glutamate released in the synaptic cleft is taken up by astrocytes to maintain glutamate homeostasis [354]. Simultaneously, astrocytes continue to take up glucose from the brain circulation [355], while glucose is converted into lactate by glycolysis and lactic acid fermentation [356]. Astrocytic lactate is then shuttled to the neighboring presynaptic neurons through astrocytic monocarboxylate transporters 1 and 4 (MCT1/4) and neuronal MCT2 [357]. This idea has been recently exemplified by adenovirus-mediated conditional knockout of neuronal MCT2 and astrocytic MCT4 in a rodent study. Conditional knockout of neuronal MCT2 and astrocytic MCT4 impair learning and memory performances in the hippocampus-dependent tasks [358]. Importantly, lactate infusion in the cerebral ventricles restores spatial learning in MCT4 knockout mice, suggesting that lactate shuttling by the astrocytes is pivotal to the direct effect of lactate, which contributes to hippocampal-dependent learning [358]. Notably, neuronal MCT2-knockout, but not astrocytic MCT4-knockout, reduces the immature neuron population in the hippocampal dentate gyrus [358], suggesting MCT2 is involved in regulating adult neurogenesis in the hippocampus. All in all, these results have supported the lactate shuttle hypothesis and how astroglial-neuron shuttle is involved in hippocampal-dependent learning and memory.

### 9.4. β-Hydroxybutyrate

β-Hydroxybutyrate is a hepatokine, which is synthesized in the liver from fat circulating in the bloodstream during fasting and exercise [359]. During diabetic ketoacidosis, increased β-oxidation and acidosis reduces mitochondrial redox state in favour of β-hydroxybutyrate biosynthesis in the liver [360]. Conversely, running exercise elevates the circulating level of β-hydroxybutyrate [361]. β-hydroxybutyrate seems to elicit both anti-inflammatory and neurotrophic effects. β-hydroxybutyrate is an endogenic NLRP3 inflammasome inhibitor [362,363], while subcutaneous administration can reduce IL-1β and TNF-α levels in the hippocampus after acute immobilization and chronic unpredictable stress [364]. Intragastric administration of β-hydroxybutyrate improves spatial learning and memory in naïve rats [365] and AD mice [366]. Another study further demonstrates that wheel running exercise increases hippocampal β-hydroxybutyrate levels in mice [367]. Intraventricular infusion of β-hydroxybutyrate mimics the exercise effect by upregulating *Bdnf* expression in the hippocampus, possibly through suppressing HDAC2/3 [367]. Glutamatergic signalling in the Schaffer-collateral path is enhanced upon acute β-hydroxybutyrate incubation [367]. Moreover, oral administration of β-hydroxybutyrate to rats for five days improves their spatial learning and memory and enhances their endurance in a treadmill test [368]. Besides, in vitro study further illustrates that β-hydroxybutyrate stimulates mitochondrial respiration and increases ATP formation in cultured cortical neurons [361]. The increased mitochondrial activity may further activate NF-κB/BDNF pathway to elicit neurogenic effect [361].

## 10. Conclusions

Emerging evidence has suggested that obesity and diabetic condition could increase the risk for neurodegeneration, and hence cognitive impairments. This review has summarized the experimental evidence demonstrating the potential influences of chronic inflammation in adipose tissue on inducing neuroinflammation and impairing neural plasticity in the brain under diabetic and obese condition. The chronic, low-grade inflammatory response in adipose tissue adversely suppresses the synthesis and secretion of anti-inflammatory adipokines, such as adiponectin. Elevated IL-1β level triggers an inflammatory response, whereas suppressed adiponectin secretion triggers insulin resistance. Aberrant peripheral secretion of adipokines in obesity and diabetes can trigger neuroinflammatory response, which in turn impairs synaptic plasticity and cognitive functioning. Mitochondria play a crucial role in maintaining neuroinflammatory response and neural plasticity in the brain. Obesity and diabetes impair mitochondrial function, that could also contribute to neuropathology, including spine loss, microglial activation, AD pathology in the brain. Physical exercise improves metabolic and brain functions. Factors including apelin, irisin, lactate, and β-hydroxybutyrate secreted from the adipose tissue, skeletal muscles, and liver are increased in response to physical exercise. These metabolic factors promote not only metabolic functions in the peripheral organs, but also maintain mitochondrial homeostasis and neural plasticity in the brain. 

In sum, convergent evidence has illustrated that obesity or diabetes triggers neuroinflammation, insulin resistance, the mitochondrial perturbation in the brain, which could impair hippocampus neuroplasticity and hence cognitive function. These findings have highlighted that neurodegeneration could be accelerated by a chronic metabolic disorder, whereas targeting some key metabolic factors that respond to physical exercise could pave the way for therapeutics and augment deficits in both metabolism and neurodegeneration. 

## Figures and Tables

**Figure 1 ijms-22-00201-f001:**
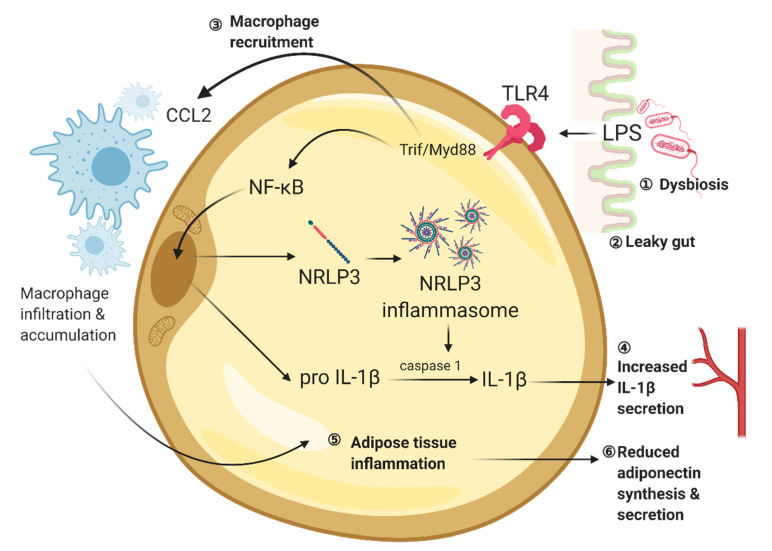
The high-fat diet promotes gut dysbiosis and adipose tissue inflammation. ① The chronic high-fat diet consumption upset the gut microbiome with an increased ratio of gram-negative bacteria to gram-positive bacteria. ② Endotoxemia reduces endothelium integrity, which allows the leakage of endotoxin, such as lipopolysaccharides. ③ LPS-induced activation of TLR4/Trif/MyD88 promotes macrophage recruitment through the secretion of CCL2. ④ Meanwhile, the activation of TLR4 also promotes NRLP3 and pro IL-1β syntheses, facilitating caspase 1-mediated IL-1β formation. ⑤ In obesity, adipose tissue macrophages further infiltrate and accumulate in the fat tissue results in chronic, low-grade inflammation, while ⑥ adiponectin synthesis and secretion are reduced. Created with BioRender.com.

**Figure 2 ijms-22-00201-f002:**
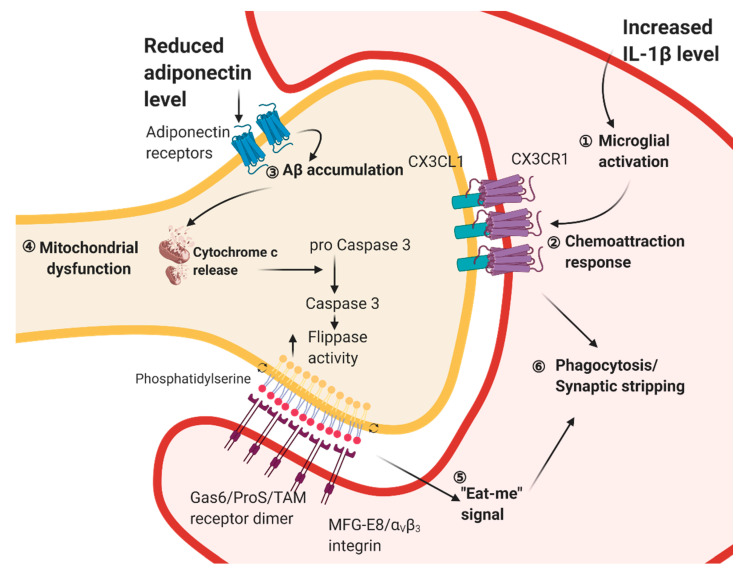
Intrinsic aberrant microglial activity and apoptotic spines can trigger synaptic stripping. ① HFD-induced hippocampal IL-1β elevation promotes the pro-inflammatory M1 phenotype in microglia. ② The aberrant microglia are attracted by CX3CL1 (Fractalkine) from the dendritic spine, which is recognized by the microglial CX3CR1. On the other hand, ③ reduced adiponectin level induces the accumulation of β-amyloid in the hippocampus. ④ Aβ accumulation causes mitochondrial dysfunction with the reduced mitochondrial membrane potential as well as the increased oxidative stress. The apoptotic mitochondrial releases cytochrome c which can activate flippase activity through caspase 3. ⑤ Phosphatidylserine located at the outer membrane of the phospholipid bilayer are served as an ‘eat-me’ signal, which is recognized by the receptors on the microglia. ⑥ Together, both aberrant microglial activity and malfunctioned neuronal mitochondria can trigger synaptic stripping by the microglia. Created with BioRender.com.

**Figure 3 ijms-22-00201-f003:**
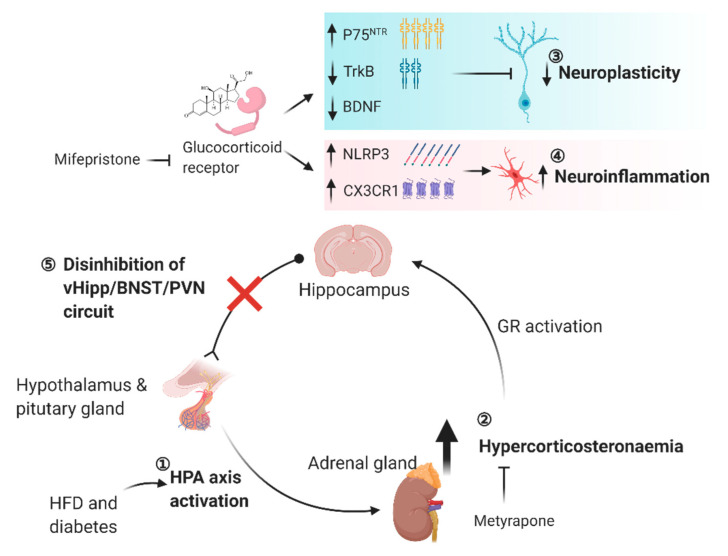
Obese and diabetic conditions disinhibit the hippocampal inhibitory control over the HPA axis. ① HPA axis hyperactivation is reported in diet-induced obese and diabetic conditions, resulting in ② increased corticosterone synthesis and secretion. Corticosterone can cross the blood-brain barrier and activates the glucocorticoid receptors in the hippocampus. The activation of the glucocorticoid receptor ③ impairs neuroplasticity and ④ promotes neuroinflammatory response. ⑤ The hippocampus is shown to inhibit the HPA activity through the vHipp/BNST/PVN neural circuit pathway, whereas dysregulated hippocampal plasticity may disinhibit the HPA axis and aggravates the diet-induced hyperactivation of the HPA axis. Created with BioRender.com.

**Figure 4 ijms-22-00201-f004:**
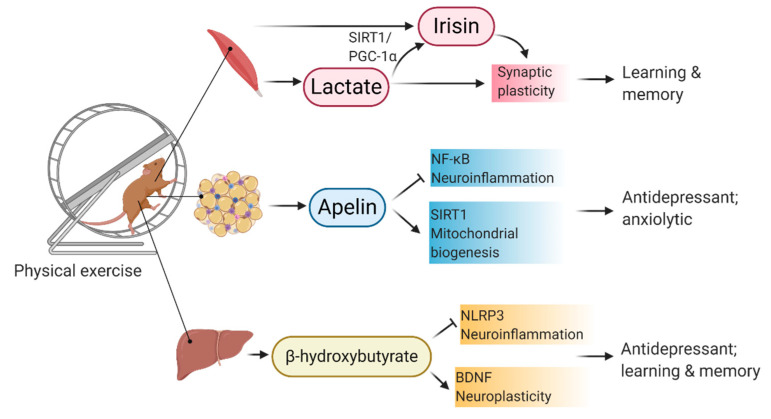
Cytokines secreted from peripheral tissue organs during exercise elicit pro-cognitive effects. The skeletal muscle secrets both lactate and irisin. In particular, both lactate and irisin can separately or concomitantly are engaged in modulating synaptic plasticity in the hippocampus. The adipose tissue secretes apelin, which reduces neuroinflammatory responses and promotes mitochondrial biogenesis in the neurons, respectively. The liver secretes β-hydroxybutyrate, a natural NLRP3 inhibitor, which can suppress the neuroinflammatory response. β-hydroxybutyrate can also promote neuroplasticity alongside with upregulating BDNF expression in the hippocampus. These exerkines are shown to be pro-cognitive by modulating both mood-related behaviours and learning and memory performance in the rodent studies. Created with BioRender.com.

## Data Availability

No new data were created or analyzed in this study. Data sharing is not applicable to this article.

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
