# Peer review of "From Obesity to Hippocampal Neurodegeneration: Pathogenesis and Non-Pharmacological Interventions"

_ijms, 2020, doi:10.3390/ijms22010201_

Round 1

Reviewer 1 Report

The review paper “From obesity to hippocampal neurodegeneration: Pathogenesis and Therapeutics” by Thomas Ho-yin Lee and Suk-yu Yau is interesting and focusses on a very timely topic. The manuscript is well written and, for most parts, easy to follow.

GENERAL COMMENTS (MINOR)

  • The authors should consider adding a few figures to increase the readability of the paper. For instance, a graphical abstract and/or other figures illustrating the mechanisms that link obesity to neurodegeneration and indication possible targets for therapies.
  • The part about “therapeutics” (which is part of the title) is largely left up to the reader to deduce from the description of the mechanisms of actions and effects. The manuscript would benefit from more explicit comments to what the authors see as the most promising therapeutic targets. This could either be a separate paragraph towards the end of the paper, or possible therapeutic targets should be pointed out more clearly in the text. Are any therapeutics under development (clinical trials)?
  • Perhaps it would be relevant to site the review paper by Cunnane et al https://pubmed.ncbi.nlm.nih.gov/32709961/ which discusses brain energy rescue in age related neurodegenerative disorders? And this review about adipokines linking obesity to dementia: https://www.ncbi.nlm.nih.gov/pmc/articles/PMC4169761/
  • In general, information about which receptors are present in the brain and that can be acted on by the adipokines and myokines are not discussed in detail. I would suggest including this information, as it may help understanding the mechanisms behind the effects of these substances. Consider adding figures to illustrate their locations and/or the mechanisms linking activation of (some of) these receptors to enhanced neuroinflammation or reduced brain function.
  • Effects of cortisol is not mentioned, even if cortisol is a known inhibitor of adult neurogenesis and enhanced cortisol levels are seen in animal models and patients with type II diabetes.
  • The authors focus on HFD, but do not mention effects of the ketogenic diet. Perhaps this should be included?
  • For completeness, the authors may consider to add a paragraph or two about the gut-brain-axis and what is known about the microbiota in obesity and diabetes.
  • In the last paragraph of the main text, the authors briefly mention lactate as a possible myokine, and site two papers that demonstrate a metabolic effect of lactate in the brain. This is now under the heading “Irisin”. I would suggest making a new paragraph where effects of lactate are discussed. The results obtained by Lev-Vachnish and colleagues are a bit hard to interpret, as the authors appear to have injected an MCT2 blocker systemically. Most MCT-blockers also affect mitochondrial uptake of pyruvate. How the inhibition of MCT2, and perhaps mitochondrial pyruvate transport, in all cells affect brain function is not easy to deduce from this paper. Does the MCT2 blocker pass the BBB? Other papers to add are these: E et al., 2013. J. Neurochem; Alvarez et al. 2014. Biomaterials. Perhaps it would even be relevant to refer to Brook’s lactate shuttle hypothesis?
    It could also be interesting to include the possibility of lactate acting via the lactate receptor HCA1 (aka HCAR1; GPR81). The presence of HCA1 in the brain was discovered simultaneously by two groups: (Bozzo et al 2013. PLoS One; Lauritzen, K. H. et al. 2014. Cereb. Cortex) and later (Hadzic et al. 2020; IJMS). HCA1 activation has been suggested to induce angiogenesis (Morland et al., 2017, Nature Comm) and to affect neuronal activity (Bozzo et al 2013. PLoS One). See also antidepressant effects of lactate (Carrad et al., 2018. Mol Psychiatry).
  • Please read carefully through the manuscript. There are some “the” missing here and there and a few extra spaces. Other than that, I did not find many misspellings (some are pointed out in my detailed comments).

SPECIFIC COMMENTS (MINOR)

P1, L34: I believe the authors mean “peripheral organs” instead of “systemic organs”? This phrase is used several places in the manuscript.

P2, L57: The authors write: “Long-term potentiation (LTP) and long-term depression (LTD) are cellular mechanisms that indicate changes in synapse plasticity” Do the authors actually mean changes in synapse plasticity or rather that LTP and LTD are mechanisms involved in synaptic plasticity?

P2, L78-79: The authors state that a reduction in the expression levels of PSD and gephyrin suggests that HFD affects both excitatory and inhibitory neurotransmitters in the hippocampus. A reduction in scaffolding proteins in the postsynaptic membrane is like not affecting the neurotransmitters per se (at least not directly), hence a more correct choice of words would perhaps be “excitatory and inhibitory synapses”.

P3, L91: What does “this study” point back to: the present study (likely not, since it is a review paper), reference 52? Please clarify.

P4, L43: “anddiabetic” should be “and diabetic”.

P4. L146: I suggest rewriting the sentence “These macrophages can take up to 40% of all cells in the adipose tissue”. The macrophages certainly don not take up 40 % of the adipocytes, as this sentence implies.  What the authors mean is probably that the macrophages constitute 40 % of the cells (as opposed to 10 % in the normal condition)?

P4. L147-149: References are lacking behind the statement “Several fat transplantation studies demonstrate that inflammatory response in adipose tissue could be linked to obesity and diabetes-induced deficits in hippocampal plasticity and cognitive functions”.

P5, L176: should “pro-inflammatory cytokine” be changed to the plural form?

P5, L198: Please remove the “i” in “….well as i adipose tissue inflammation”

P5, L198: I do not understand the sentence “These perturbations further promote microglial activation as impair hippocampal plasticity”. Please revise.

P5, L199-200: Please clarify what is meant by “Studies have shown that dietary withdrawal is an ineffective non-pharmacological intervention for improving learning and memory deficits (96, 97)”. What do the authors mean by “dietary withdrawal”? The two sited papers demonstrate effects of dietary interventions, and the next couple of sentences also suggest that the authors mean that there is an effect of diets on measures of learning and memory.

P5, L206: Please use the more common term “synaptic terminals” instead of “synaptic termini”.

P6, L237: Why is “expressions” in plural?

P6, 238-239: It would be more correct to present phospholipid scramblase and flippase as two families of enzymes that catalyze the translocation of phospholipids. Furthermore, I suggest to include “between the two monolayers of the cell membrane”, for clarity.

P7. L267: Stating that adipokines mediate (all of) the pro-cognitive effects of physical exercise seems like an over-simplification. What about myokines? The authors also present some factors other than adipokines that affect cognition. Hence, the above-mentioned sentence should be revised.

P7, L290: 2.1. should be 4.1.

P8, L298: I suggest changing “were discussed” to “are discussed below”.

P8, L314: 2.2. should be 4.2.

P9, L339-341: The review paper, by definition, does not provide experimental support or give new insight (as no new experiments were done, the insight -in theory- is already available). The authors should consider rewriting this sentence to highlight the fact that the review paper summarizes the evidence.

P9, L348 “Obese and diabetic dysregulates” should be changes to “Obesity and diabetes leads to a dysregulation of..” or “Obesity and diabetic dysregulation of…”

P9, L350-351: The statement “Factors including apelin and irisin, secreted from the adipose tissues are increased in response to physical exercise” is not correct as irisin is mainly released from the exercising muscles, not primarily form the adipose tissue.

P9, L352: “promoting” should be changes to “promote”.

Author Response

Reviewer 1:

Thank you so much for reviewer’s positive comments. We have improved the contents as suggested by the reviewer accordingly. Here are the point-to-point response to the major suggestions.

The part about “therapeutics” (which is part of the title) is largely left up to the reader to deduce from the description of the mechanisms of actions and effects. The manuscript would benefit from more explicit comments to what the authors see as the most promising therapeutic targets. This could either be a separate paragraph towards the end of the paper, or possible therapeutic targets should be pointed out more clearly in the text. Are any therapeutics under development (clinical trials)?

Response: Thank you for your comments. We have changed the title from therapeutics to non-pharmacological interventions, and have included ketogenic diet (from Line 478 – 550), and caloric restriction (Line 551 – 558) as non-pharmacological interventions with discussion on the potential therapeutic targets.

Line 478 - 550:

“7. High-fats and low-carb ketogenic diet elicits neuroprotective effects by promoting mitochondrial dynamics and reducing oxidative stress

There are misconceptions that high fat and dietary carbohydrate intake is directly linked to metabolic syndrome. Instead, high-fat diet elicits obesogenic effect when the caloric balance is upset by a high caloric intake versus a low energy expenditure, for example, physical inactivity.   Counterintuitively, a high-fats, low-carbohydrate ketogenic diet (up to 90% fats in rodent studies) has been employed as a non-pharmacological diet intervention for controlling weight gain, glycemic index, and lipid content (248) and prevention for the recurrences of epilepsy (249).

Adopting a high-fat and very low-carbohydrates diet mimics the metabolic states of fasting and prolonged physical exercise, and elevates the levels of ketone bodies, known as physiologic ketosis (250). When glucose reserve is depleted, lipids become the major source for ATP biosynthesis. Triacylglyceride is catalyzed into acetyl-CoA through β-oxidation, which then metabolized into ketone bodies, including acetone and β-hydroxybutyrate. β-hydroxybutyrate is the substrate for ATP biosynthesis. Ketosis in physiological state is different from diabetic ketoacidosis. Diabetic ketoacidosis, on the other hand, is a life-threatening complication of diabetes mainly due to the massive breakdown of lipids in adipose tissue induced by insulin deficit (251). The high free fatty acid in the circulation accounts for approximately 80% of energy through β-oxidation, leading to the accumulation of ketone bodies (20–25 mM) at a level much higher than that during fasting or consuming a ketogenic diet (4-6 mM) (252-257).

Ketogenic diet elicits anti-inflammatory effect on adipose tissues under short-term consumption. Short-term exposure to a ketogenic diet (89.5% fats; 0.1% carbohydrates) for one week promotes ketogenesis with increased circulating β-hydroxybutyrate level in fed and fasting states (258). Short term ketogenic exposure reduces the population of pro-inflammatory macrophages in the adipose tissue and down-regulates the Nlrp3 and Il1b expressions in the adipose-resident immune cells. In another study, it is also reported that four-week ketogenic diet (93.4% fat, 1.8% carbohydrate) down-regulates the expressions of inflammatory markers, such as Il6, Tnf, and Nlrp3, in the epidydimal white adipose tissue (259). On the contrary, chronic exposure to a ketogenic diet (89.5% fats; 0.1% carbohydrates) for four months imposes detrimental effect on adipose tissue metabolism (258). Chronic exposure promotes body weight gain and hyperglycemia (258). Macrophage population is significantly increased in association with up-regulated inflammatory marker expressions, including Mcp1, Tnfa, and Il1b, in the epididymal fat tissues (258). Still, the circulating β-hydroxybutyrate level remains high after four months of consuming a ketogenic diet. After that, an independent study attempts to harness the beneficial effect of ketogenesis by adopting a cyclic ketogenic diet paradigm. From 12-month-old onwards, naïve mice receive an alternating ketogenic diet (90% fats; 0% carbohydrates) and standard chow (13% fats; 77% carbohydrates) weekly until 36 months old (260). Cyclic ketogenic diet elevates blood β-hydroxybutyrate levels but does not induce body weight gain. Aged mice have better memory recall in place avoidance test (at 22 to 24 months old) and object recognition task (at 28 to 30 months old) on a cyclic ketogenic diet than receiving standard chow. Also, the cyclic ketogenic diet reduces early mortality in naïve mice at 12 to 24 months old.

The dynamics of the NAD+/NADH ratio implies mitochondrial activity with increased mitochondrial fission increases the NAD+ levels. The two-week ketogenic diet (6:1 fats-to-carbohydrates and proteins ratio) elevates hippocampal NAD+/NADH ratio in 11 to 14-week-old rats (261). In vitro study suggests that β-hydroxybutyrate may contribute to the increase in NAD+/NADH ratio (262). Poly-ADP ribose polymerase-1 (PARP1) serves as a DNA damage sensor and participates in DNA repair (263). While NAD+ is a substrate for PARP1, DNA damage triggers PARP1-induced NAD+ depletion (264). The ketogenic diet, on the other hand, reduces hippocampal PARP-1 and 8-hydroxy-2’-deoxyguanosine levels, suggesting a potential reduction in DNA damage or enhanced DNA repair activity. SIRT-1 activity reduces cell death and inflammation (265, 266) and promotes neuronal survival (267-269), while ketogenic diet promotes the activities of nuclear sirtuins as well as the hippocampal Sirt1 expression (261). Altogether, the findings have suggested that the ketogenic diet could exert neuroprotective by promoting mitochondrial dynamics and reducing oxidative stress in neurons, which may serve as a dietary intervention to prevent cognitive decline. 

Regardless, the effect of a ketogenic diet (84% fats; 0% carbohydrates) presents an adverse effect in the transgenic mice expressing a mutated mitochondrial DNA repair enzyme (mutUNG1) selectively in forebrain neurons (270). The continuous ketogenic diet for two to four months reduces hippocampal size in mutUNG1 mice. Immunofluorescence shows a reduction in neuronal marker expression and an elevation of astroglia marker expression in the hippocampus upon a ketogenic diet in mutUNG mice. The increased SOD2/VDAC1 ratio as well the increased SIRT1 and FIS1 expressions in the hippocampus suggests ketogenic diet activates mitochondrial antioxidant defenses and mitochondrial fission in mutUNG mice. Alterations in mitochondrial morphology, as well as the distribution of mitochondria within a neuron, are observed. For example, ketogenic diet induces swelling of mitochondria in mutUNG mice, whereas mitochondria are accumulated in the soma with reduced mitochondrial density in the presynaptic terminal of the pyramidal neurons in the hippocampal CA1. Lastly, electronic microscopy illustrates that ketogenic diet reduces the density of GluN2A and 2B subunits while increases the density of GABAA α1 subunits in the hippocampal CA1 of mutUNG1 mice. This study calls for rectification on whether a ketogenic diet is a proper way to augment neurodegeneration and cognitive impairments in association with severe mitochondrial dysfunction.

It is noteworthy that the peripheral metabolic, neuroprotective, and pro-cognitive effects of the ketogenic diet have called for substantial investigations. The ketogenic response is effective under low blood glucose and low insulin levels. Hyperglycemia and insulin resistance are hallmarks of diabetic patients, and these conditions are often shared in AD patients and older adults.”

Line 551 - 558:

“8. SIRT1/SIRT3 axis potentiates the pro-cognitive effects of caloric restriction and intermittent fasting

Sirtuins (SIRT) are NAD+-dependent deacetylases (271). SIRT1 is a cytosolic and nuclear sirtuin modulates transcription factors through histone deacetylation (265, 272-275). SIRT3 is a mitochondrial sirtuin regulating β-oxidation (276). SIRT1/SIRT3 axis aims to promote energy storage intracellularly and to resist oxidative stress (277, 278).

In vivo studies illustrate that SIRT1 is a mediator of caloric restriction-induced mitochondrial biogenesis in mice (279-281), while SIRT1 knockout abrogates the metabolic benefits of caloric restriction (274, 282). In the hippocampus, 23 weeks of HFD reduces the expressions of Ppargc1a, Ppp1cb, Reln and Sirt1 (283). Increased DNA methylation and decreased DNA hydroxymethylation are observed in the promoter region of Sirt1, implicating a diet-induced epigenetic modulation of Sirt1 (283). On the contrary, caloric restriction promotes learning and memory performance as well as hippocampal synaptic plasticity by upregulating signaling transducers involved in mitochondrial biogenesis (284), including CREB phosphorylation and increased expression of SIRT1, PGC-1α, nitric oxide synthase, and phosphoenolpyruvate carboxykinase in the hippocampus (284). The activation of a mitochondrial biogenesis signaling pathway is associated with better learning and memory performance in the novel object recognition task and enhanced LTP formation in the Schaffer-collateral path in a neuronal-specific CREB-dependent manner (284).

SIRT3 is one of the SIRT1 deacetylation targets. HFD-induced obesity and aging obliterate SIRT1-mediated deacetylation of SIRT3, which reduces SIRT3 activity and stability in the mouse model (285). Intermittent caloric restriction, as short as one week to one month, promotes the hippocampal SIRT3 expression (286). Besides, the anxiolytic and pro-cognitive effects of intermittent caloric restriction are abolished when SIRT3 is selectively knockout in the hippocampal neurons. Intermittent caloric restriction further impairs LTP formation in the Schaffer-collateral path of the SIRT3-deficient mice. Significantly, intermittent fasting reduces seizure incidence, restores hippocampal-dependent spatial memory performance, as well as rescues LTP deficits in AppNL-G-F mice of AD.

SIRT3 can deacetylate LKB1, resulting in AMPK activation and mitochondrial biogenesis (287). Four-week intermittent fasting activates AMPK/PGC-1a/mTORc/COX4/ND-1 signaling cascade, suggesting that intermittent fasting promotes mitochondrial biogenesis in the hippocampus. The increased mitochondrial DNA content further evidences this result in the hippocampus in db/db mice (288). Essentially, intermittent fasting also exerts pro-cognitive and neurotrophic effects by restoring diabetes-induced hippocampal-dependent spatial memory deficits and the morphology of postsynaptic density of the neurons in the hippocampal CA1 region in addition to the activation of BDNF/pCREB/PSD-95 signaling cascade (288).

In sum, SIRT1/SIRT3 axis is one of the molecular targets of caloric restriction-induced pro-cognitive and neurotrophic effects, partly due to its functional role in regulating mitochondrial biogenesis.”

Perhaps it would be relevant to site the review paper by Cunnane et al https://pubmed.ncbi.nlm.nih.gov/32709961/ which discusses brain energy rescue in age related neurodegenerative disorders? And this review about adipokines linking obesity to dementia: https://www.ncbi.nlm.nih.gov/pmc/articles/PMC4169761/

Response: Thank you so much for your suggestions. We have included Cunnane et al (line 39 – 40) and Arnoldussen et al ideas (line 278 - 280) in the revised manuscript.

Line 64 - 67: “β-amyloid accumulation is one of the pathologies found in the AD brains. In the form of amylin or islet amyloid polypeptide accumulation, amyloidosis is also observed in the pancreas under diabetic condition (30).”

Line 319 - 320: “Restoring or enhancing brain energetics has been proposed to be a potential therapeutic approach to halt neurodegenerative diseases of aging (196).”

In general, information about which receptors are present in the brain and that can be acted on by the adipokines and myokines are not discussed in detail. I would suggest including this information, as it may help understanding the mechanisms behind the effects of these substances. Consider adding figures to illustrate their locations and/or the mechanisms linking activation of (some of) these receptors to enhanced neuroinflammation or reduced brain function.

Response: Thank you for tour suggestion. We have included four new schematics to illustrate the ideas presented in this review.

Effects of cortisol is not mentioned, even if cortisol is a known inhibitor of adult neurogenesis and enhanced cortisol levels are seen in animal models and patients with type II diabetes.

Response: The role of diabetes-induced corticosterone elevation has been deeply discussed in the new Section 6. Obese and diabetic conditions induce feedback inhibition of hippocampal control over HPA axis (Line 398 – 468) as below:

“Line 398 - 468: 6. Obese and diabetic conditions inhibit hippocampal feedback control of the HPA axis

An epidemiological study has reported that type 2 diabetic individuals have a higher prevalence of hypercortisolemia (226) while hypercorticosteronemia is also observed in diet-induced obese and genetically diabetic rodent models (227, 228). The increased activity in the hypothalamic-pituitary-adrenal axis (HPA) is a major cause of elevated cortisol or corticosterone secretion from the adrenal gland (229). Lesion studies have illustrated that the hippocampus exerts efferent inhibitory control of the HPA axis. It is reported that total hippocampectomy elevates the expression of corticotropin-releasing hormone in the hypothalamic paraventricular nucleus (PVN) (230), whereas glucocorticoid secretion is markedly increased in rats bearing lesion in the ventral hippocampus (105). Besides, recent circuit studies have revealed that the glutamatergic projections from ventral hippocampus synapses on the bed nucleus of striatum terminalis (BNST), whereby the BNST GABAergic projection elicits an inhibitory control over PVN (125, 231). It is reasonable to speculate that obese- or diabetes-induced hippocampal impairment may abolish the efferent inhibitory control over the HPA axis leading to hypercorticism. The presence of glucocorticoid receptors (232) and the ability that corticosterone can cross the blood-brain barrier (233) further implicate that the inhibitory action of hippocampus on HPA axis involves feedback inhibition. This is supported by the fact that deletion of glucocorticoid receptors in the corticolimbic forebrain, including the hippocampal region, prolongs HPA axis activation (234, 235).

Acute exposure of high-fat diet for three days is sufficient to elevate corticosterone levels in rats and triggers neuroinflammatory responses with increased inflammasome-associated NLRP3 and CX3CR1 expressions in the hippocampus (103). Acute HFD exposure leads to a higher vulnerability to a lipopolysaccharide-triggered inflammatory response with much higher hippocampal IL-1β and IL-6 expressions (103). Independent of hypercorticosteronemia, systemic administration of a blood-brain barrier permeable glucocorticoid receptor antagonist: mifepristone (236) (also known as RU-486; 50 mg/kg s.c., two doses in three days) suppresses the expressions of neuroinflammatory markers in the hippocampus and attenuates lipopolysaccharide-induced neuroinflammation (103). In addition to the pro-inflammatory effect of corticosterone, corticosterone impairs hippocampal plasticity, abrogates learning and memory performance, and reduces BDNF expression (107, 109, 237-239). Conversely, mifepristone administration not only rescues memory deficits in an object recognition task but also restores LTP deficit in the Schaffer-collateral path in three-week-old juvenile rats exposed to the high-fat diet for 7 to 9 days (240). Potentially, the high-fat diet-induced corticosterone surge impairs hippocampal function and plasticity, which in turn weakens the efferent inhibitory control over the HPA axis, where hypercorticosteronemia is observed in obesity and diabetes.

However, normalizing corticosterone level in the circulation or blocking the central action of corticosterone can be a remedy to hypercorticosteronemia-induced neuroinflammation and impaired hippocampal plasticity in obesity and diabetes. Systemic administration of metyrapone reduces circulating corticosterone level by suppressing corticosterone synthesis (241, 242). In dependent of insulin sensitivity in the hippocampus, metyrapone administration in five-week-old db/db mice (100 mg/kg i.p., three weeks) reduces microgliosis and suppresses the expressions of IL-1β and TNFα in the hippocampus (243). In vitro study has further illustrated that metyrapone administration reduces the pro-inflammatory M1 microglia population in the hippocampus of db/db mice as well as reduces microglial sensitivity against lipopolysaccharide assault in primary microglia culture (243). These findings emphasize that the direct pro-inflammatory effect of corticosterone in the hippocampus regardless of a short-term high-fat diet exposure context or a genetically diabetic background. In db/db and streptozotocin-induced diabetic mice, adenectomy with concomitant corticosterone replacement restores cell proliferation and LTP formation in the hippocampus as well as improves learning and memory performance (44). These restorations are linked to the reduction in circulating corticosterone levels and are independent of diabetes-induced hyperglycaemia (44). In situ hybridization further reveals that adenectomy and corticosterone replacement restores BDNF-TrkB expressions in the hippocampal dentate region of db/db mice (244). Moreover, suppression of corticosterone synthesis by systemic metyrapone administration in db/db mice rescues spatial memory deficits in concurrent with restored LTP formation and spine density of the granule cells in the hippocampus (228). Systemic metyrapone administration fails to rescue the corticosterone-induced memory impairment, LTP deficits, and spine loss when corticosterone is locally infused in the hippocampus (228), suggesting direct central action of corticosterone.

Corticosterone fails to induce spatial memory and synaptic impairments in db/db mice when glucocorticoid receptors are knocked down in the hippocampus (228), suggesting the potential role of glucocorticoid receptor signaling in the corticosterone action. Corticosterone impairs the hippocampal BDNF signaling pathway in db/db mice by two means. Firstly, the activation of glucocorticoid receptor transcriptionally suppresses the hippocampal BDNF expression. Concomitantly, the expressions of BDNF receptors are altered in db/db mice. While BDNF has a higher affinity for TrkB receptor, which promotes neural plasticity (245), it has a lower affinity for the P75 neurotrophin receptor (P75NTR) (246), which constrains plasticity (247). TrkB expression is down-regulated, and P75NTR expression is up-regulated in the hippocampus of db/db mice, while pharmacological inhibition of corticosterone synthesis by metyrapone can re-activate the BDNF/TrkB cascade and bypass the P75NTR pathway. These studies demonstrate the complexity of the negative feedback mechanism of the hippocampus-HPA axis in response to obesity and diabetes-induced hypercorticosteronemia, in which the action of corticosterone on activating inflammasome-associated response and suppressing BDNF signaling induces neuroinflammation and plasticity deficits in the hippocampus (Figure 3).”

The authors focus on HFD, but do not mention effects of the ketogenic diet. Perhaps this should be included?

Response: We have included the new Section 7 to discuss about the neuroprotective effects of ketogenic and its mechanisms (Line 418 - 488) as below:

Line 478 - 550:

“7. High-fats and low-carb ketogenic diet elicits neuroprotective effects by promoting mitochondrial dynamics and reducing oxidative stress

There are misconceptions that high fat and dietary carbohydrate intake is directly linked to metabolic syndrome. Instead, high-fat diet elicits obesogenic effect when the caloric balance is upset by a high caloric intake versus a low energy expenditure, for example, physical inactivity.   Counterintuitively, a high-fats, low-carbohydrate ketogenic diet (up to 90% fats in rodent studies) has been employed as a non-pharmacological diet intervention for controlling weight gain, glycemic index, and lipid content (248) and prevention for the recurrences of epilepsy (249).

Adopting a high-fat and very low-carbohydrates diet mimics the metabolic states of fasting and prolonged physical exercise, and elevates the levels of ketone bodies, known as physiologic ketosis (250). When glucose reserve is depleted, lipids become the major source for ATP biosynthesis. Triacylglyceride is catalyzed into acetyl-CoA through β-oxidation, which then metabolized into ketone bodies, including acetone and β-hydroxybutyrate. β-hydroxybutyrate is the substrate for ATP biosynthesis. Ketosis in physiological state is different from diabetic ketoacidosis. Diabetic ketoacidosis, on the other hand, is a life-threatening complication of diabetes mainly due to the massive breakdown of lipids in adipose tissue induced by insulin deficit (251). The high free fatty acid in the circulation accounts for approximately 80% of energy through β-oxidation, leading to the accumulation of ketone bodies (20–25 mM) at a level much higher than that during fasting or consuming a ketogenic diet (4-6 mM) (252-257).

Ketogenic diet elicits anti-inflammatory effect on adipose tissues under short-term consumption. Short-term exposure to a ketogenic diet (89.5% fats; 0.1% carbohydrates) for one week promotes ketogenesis with increased circulating β-hydroxybutyrate level in fed and fasting states (258). Short term ketogenic exposure reduces the population of pro-inflammatory macrophages in the adipose tissue and down-regulates the Nlrp3 and Il1b expressions in the adipose-resident immune cells. In another study, it is also reported that four-week ketogenic diet (93.4% fat, 1.8% carbohydrate) down-regulates the expressions of inflammatory markers, such as Il6, Tnf, and Nlrp3, in the epidydimal white adipose tissue (259). On the contrary, chronic exposure to a ketogenic diet (89.5% fats; 0.1% carbohydrates) for four months imposes detrimental effect on adipose tissue metabolism (258). Chronic exposure promotes body weight gain and hyperglycemia (258). Macrophage population is significantly increased in association with up-regulated inflammatory marker expressions, including Mcp1, Tnfa, and Il1b, in the epididymal fat tissues (258). Still, the circulating β-hydroxybutyrate level remains high after four months of consuming a ketogenic diet. After that, an independent study attempts to harness the beneficial effect of ketogenesis by adopting a cyclic ketogenic diet paradigm. From 12-month-old onwards, naïve mice receive an alternating ketogenic diet (90% fats; 0% carbohydrates) and standard chow (13% fats; 77% carbohydrates) weekly until 36 months old (260). Cyclic ketogenic diet elevates blood β-hydroxybutyrate levels but does not induce body weight gain. Aged mice have better memory recall in place avoidance test (at 22 to 24 months old) and object recognition task (at 28 to 30 months old) on a cyclic ketogenic diet than receiving standard chow. Also, the cyclic ketogenic diet reduces early mortality in naïve mice at 12 to 24 months old.

The dynamics of the NAD+/NADH ratio implies mitochondrial activity with increased mitochondrial fission increases the NAD+ levels. The two-week ketogenic diet (6:1 fats-to-carbohydrates and proteins ratio) elevates hippocampal NAD+/NADH ratio in 11 to 14-week-old rats (261). In vitro study suggests that β-hydroxybutyrate may contribute to the increase in NAD+/NADH ratio (262). Poly-ADP ribose polymerase-1 (PARP1) serves as a DNA damage sensor and participates in DNA repair (263). While NAD+ is a substrate for PARP1, DNA damage triggers PARP1-induced NAD+ depletion (264). The ketogenic diet, on the other hand, reduces hippocampal PARP-1 and 8-hydroxy-2’-deoxyguanosine levels, suggesting a potential reduction in DNA damage or enhanced DNA repair activity. SIRT-1 activity reduces cell death and inflammation (265, 266) and promotes neuronal survival (267-269), while ketogenic diet promotes the activities of nuclear sirtuins as well as the hippocampal Sirt1 expression (261). Altogether, the findings have suggested that the ketogenic diet could exert neuroprotective by promoting mitochondrial dynamics and reducing oxidative stress in neurons, which may serve as a dietary intervention to prevent cognitive decline. 

Regardless, the effect of a ketogenic diet (84% fats; 0% carbohydrates) presents an adverse effect in the transgenic mice expressing a mutated mitochondrial DNA repair enzyme (mutUNG1) selectively in forebrain neurons (270). The continuous ketogenic diet for two to four months reduces hippocampal size in mutUNG1 mice. Immunofluorescence shows a reduction in neuronal marker expression and an elevation of astroglia marker expression in the hippocampus upon a ketogenic diet in mutUNG mice. The increased SOD2/VDAC1 ratio as well the increased SIRT1 and FIS1 expressions in the hippocampus suggests ketogenic diet activates mitochondrial antioxidant defenses and mitochondrial fission in mutUNG mice. Alterations in mitochondrial morphology, as well as the distribution of mitochondria within a neuron, are observed. For example, ketogenic diet induces swelling of mitochondria in mutUNG mice, whereas mitochondria are accumulated in the soma with reduced mitochondrial density in the presynaptic terminal of the pyramidal neurons in the hippocampal CA1. Lastly, electronic microscopy illustrates that ketogenic diet reduces the density of GluN2A and 2B subunits while increases the density of GABAA α1 subunits in the hippocampal CA1 of mutUNG1 mice. This study calls for rectification on whether a ketogenic diet is a proper way to augment neurodegeneration and cognitive impairments in association with severe mitochondrial dysfunction.

It is noteworthy that the peripheral metabolic, neuroprotective, and pro-cognitive effects of the ketogenic diet have called for substantial investigations. The ketogenic response is effective under low blood glucose and low insulin levels. Hyperglycemia and insulin resistance are hallmarks of diabetic patients, and these conditions are often shared in AD patients and older adults.”

Line 710 - 728:

9.4. β-hydroxybutyrate

β-hydroxybutyrate is a hepatokine, which is synthesized in the liver from fat circulating in the bloodstream during fasting and exercise (360). During diabetic ketoacidosis, increased β-oxidation and acidosis reduces mitochondrial redox state in favor of β-hydroxybutyrate biosynthesis in the liver (361). Conversely, running exercise elevates the circulating level of β-hydroxybutyrate (362). β-hydroxybutyrate seems to elicit both anti-inflammatory and neurotrophic effects. β-hydroxybutyrate is an endogenic NLRP3 inflammasome inhibitor (363, 364), while subcutaneous administration can reduce IL-1β and TNF-α levels in the hippocampus after acute immobilization and chronic unpredictable stress (365). Intragastric administration of β-hydroxybutyrate improves spatial learning and memory in naïve rats (366) and AD mice (367). Another study further demonstrates that wheel running exercise increases hippocampal β-hydroxybutyrate levels in mice (368). Intraventricular infusion of β-hydroxybutyrate mimics the exercise effect by upregulating Bdnf expression in the hippocampus, possibly through suppressing HDAC2/3 (368). Glutamatergic signaling in the Schaffer-collateral path is enhanced upon acute β-hydroxybutyrate incubation (368). Moreover, oral administration of β-hydroxybutyrate to rats for five days improves their spatial learning and memory and enhances their endurance in a treadmill test (369). Besides, in vitro study further illustrates that β-hydroxybutyrate stimulates mitochondrial respiration and increases ATP formation in cultured cortical neurons (362). The increased mitochondrial activity may further activate NF-κB/BDNF pathway to elicit neurogenic effect (362).”

For completeness, the authors may consider to add a paragraph or two about the gut-brain-axis and what is known about the microbiota in obesity and diabetes.

Response: Thanks a lot for the suggestion. We have discussed the contents in the new Section 2, line 145 – 193, and line 647 – 665 as below:

Line 179 - 231:

“2. Gut is a potential origin of chronic, low-grade inflammation in obesity and diabetes

Chronic, low-grade inflammation in the adipose tissue is a characteristic of diet-induced obesity (151). The critical effector that triggers diet-induced the adipose tissue inflammation is often masked. This is because metabolic and immunological complications have arisen from multiple organs in obesity, or when diabetes is diagnosed. Studies have suggested that the gut microbiome is inextricably linked to obesity. A pioneering animal study has reported that naïve recipients harboring the gut microbiota from ob/ob mice have increased percentage body fat, increased energy consumption, as well as increased acetate and butyrate concentrations in the fecal samples (152). This finding is echoed by a later study investigating the effect on adiposity by inoculating the microbiota from pairs of human twins, of whom one twin is obese, and the other is lean, in the germ-free mice. Upon the transplantation of microbiome, mice consume a low-fat (4%) and high-plant polysaccharides diet. However, mice become obese after human microbiome transplantation from the obese twin, whereas mice harboring the human microbiome from the lean twin remains lean.  When respective mice harboring the lean and the obese microbiota are co-housed, both mice are resistant to obesity. The study further reports that the Bacteroidetes species in the gut microbiota from lean humans can resist the obese microbiota invasion (153). These studies suggest that gut can be an origin of obesity.

High-fat diet induces endotoxemia with increased circulating lipopolysaccharides (154). Lipopolysaccharide is the main component of gram-negative bacteria outer membrane, constituting a vast endotoxin reservoir in the gut. High-fat diet alters the ratio of gram-negative to gram-positive bacteria in the gut microbiome with an elevated composition of gram-negative bacteria (154, 155). Concomitantly, both dietary fat and intestinal dysbiosis reduce the integrity of the intestinal lumen (156, 157). The increased gut barrier permeability, also known as the leaky gut, results in the leakage of endotoxin (154, 158). The excess entry of gram-negative bacteria-derived lipopolysaccharides into the circulation results in endotoxemia and systemic inflammation (156, 159-161).

Visceral adipose tissue is one of the target sites where lipopolysaccharides induce inflammation. 11-week lard diet increases the circulating lipopolysaccharide level and increases the expression levels of toll-like receptor 2 (TLR2) and toll-like receptor 4 (TLR4) when compared to mice receiving 11-week fish oil diet. Lipopolysaccharide binds to TLR4 on adipocyte, which in turn activates the Trif/MyD88/CCL2 signaling pathway (162-164). TRIF and Myd-88 are TLR adaptor molecules (165). Chemokine CCL2 is a mediator of macrophage accumulation in white adipose tissue in obesity (166-168). Interestingly, mice lacking TRIF and Myd-88 are protected from lard diet-induced body weight gain and white adipose tissue inflammation. Trif-knockout and Myd88-knockout also prevent lard diet-induced CCL2 expression and inflammation in white adipose tissue as well as body weight gain. TLR4 recruits TRIF and Myd-88, which promotes the expression of CCL2 in adipose tissue. The obesity-induced elevation of CCL2 level further recruits macrophage in white adipose tissue (169) (Figure 1).

The condition of the gut microbiome is shown to affect hippocampal plasticity. Young microbiome-free recipients harboring the gut microbiota from the old mice promotes hippocampal neurogenesis and longevity. Conversely, microbiome-depleted recipient transplanted with fecal microbiota of high-fat diet-fed donor presents greater anxiety-like behavior, which is accompanied by the increased expressions of lymphocyte and microglial marker in plasma and whole brain (170). Further evidence shows that fecal microbiota transplant from 24-month-old donor mice impairs spatial learning and memory performance and novel object recognition in the young recipient (171). These findings suggest that gut dysbiosis induces cognitive impairment. It is also possible that the gut has direct communication with the hippocampus (172). Circuit study reveals a more direct connection between the gut-hippocampus axis. The medial nucleus tractus solitarius (mNTS) receives gut vagal sensory input, whereas the mNTS connects the dorsal CA3 region of the hippocampus through the medial septum (173). Both non-selective subdiaphragmatic vagotomy, which eliminates all gastrointestinal vagal afferents and efferents, as well as selective gastrointestinal vagal deafferentation by injecting saporin-conjugated cholecystokinin into the nodose ganglia impairs spatial memory and contextual episodic memory, which are accompanied by reduced hippocampal BDNF and doublecortin levels (173).”

In the last paragraph of the main text, the authors briefly mention lactate as a possible myokine, and site two papers that demonstrate a metabolic effect of lactate in the brain. This is now under the heading “Irisin”. I would suggest making a new paragraph where effects of lactate are discussed. The results obtained by Lev-Vachnish and colleagues are a bit hard to interpret, as the authors appear to have injected an MCT2 blocker systemically. Most MCT-blockers also affect mitochondrial uptake of pyruvate. How the inhibition of MCT2, and perhaps mitochondrial pyruvate transport, in all cells affect brain function is not easy to deduce from this paper. Does the MCT2 blocker pass the BBB? Other papers to add are these: E et al., 2013. J. Neurochem; Alvarez et al. 2014. Biomaterials. Perhaps it would even be relevant to refer to Brook’s lactate shuttle hypothesis?

Response: We have extended our discussion in the new sub-section 9.3 Lactate (Line 663 - 709) and included the articles as suggested as below:

Line 663 – 685:

“9.3. Lactate

Lactate is another myokine that is released from muscle in response to physical exercise. Lactate can cross the blood-brain barrier via endothelial monocarboxylate transporters (339, 340). The lactate action could be elicited by activating the lactate receptor HCAR1 in the brain. HCAR1 is present in the fibroblasts and ependymal cells in the brain (341). HCAR1 mediates the lactate-induced neuronal excitability in primary cortical neurons (342) as well as high-intensity exercise-induced or lactate-induced angiogenesis in the brain (343). Interestingly, systemic lactate administration partially mimics the wheel running effect on modulating the gene expression that regulates mitochondrial biogenesis in the liver and the whole brain. Lactate administration mimics the effect of exercise by upregulating the gene expressions of PGC-1α but downregulating the expressions of PGC-1β and NRF-1 in the liver. Besides, both lactate administration and exercise increase the PGC-1 related co-activator (PRC)/vascular endothelial growth factor A (VEGFA) expression in the brain. Of note, exercise increases the mitochondrial DNA copy number in the brain and reduces TNFα expression but not after lactate administration (340). Other pro-cognitive effects, including anti-depressant effects, are illustrated upon single-dose of lactate administration in corticosterone-induced stressed mice as well as after chronic lactate administration in naïve mice (344).

An earlier study has demonstrated the involvement of lactate in promoting neurogenesis and vascularization in the postnatal mouse brain (345). Extracellular electrophysiological recordings have further revealed that lactate promotes synaptic plasticity at the Schaffer-collateral path (346). In the CA3 pyramidal neurons, lactate acts on the postsynaptic lactate receptor and may facilitate AMPAR insertion in a PI3K-dependent manner, which is evidenced by increased intrinsic excitability of CA3 pyramidal neurons (346). Potentially, lactate promotes NMDAR-mediated calcium ion influx through PI3K/PKC phosphorylation, as illustrated by an increased EPSP-spike coupling (346).”

It could also be interesting to include the possibility of lactate acting via the lactate receptor HCA1 (aka HCAR1; GPR81). The presence of HCA1 in the brain was discovered simultaneously by two groups: (Bozzo et al 2013. PLoS One; Lauritzen, K. H. et al. 2014. Cereb. Cortex) and later (Hadzic et al. 2020; IJMS). HCA1 activation has been suggested to induce angiogenesis (Morland et al., 2017, Nature Comm) and to affect neuronal activity (Bozzo et al 2013. PLoS One). See also antidepressant effects of lactate (Carrad et al., 2018. Mol Psychiatry).

Response: We have included the articles suggested and have extended our discussion in the new sub-section 9.3 Lactate (Line 686 - 709) as below: “Lactate was known as a by-product of anaerobic respiration, but it is also a source of energy, a gluconeogenic precursor, and a signaling molecule (347). Since then, the concept of lactate shuttle has been raised with two highlights. First, lactate can be exported and consumed by the neighboring cells through lactate receptors (paracrine route) or by a distal organ through circulation (endocrine route) (347). Second, the role of lactates shifts from glycolytic to oxidative in the recipient tissues and organs (347). The exercise-induced lactate/MCT/SIRT1/PGC-1α/FNDC5/BDNF signaling transduction has exemplified the endocrine perspective of lactate shuttle (336). Also, several reviews have emphasized the lactate shuttle involving in synaptic plasticity and astroglial-neuronal metabolism (348-352). Lactate enters the presynaptic neurons through MCT2 (353). In the presynaptic neurons, lactate plays an oxidative role in fueling synaptic plasticity (354). During synaptic activation, glutamate released in the synaptic cleft is taken up by astrocytes to maintain glutamate homeostasis (355). Simultaneously, astrocytes continue to take up glucose from the brain circulation (356), while glucose is converted into lactate by glycolysis and lactic acid fermentation (357). Astrocytic lactate is then shuttled to the neighboring presynaptic neurons through astrocytic MCT1/4 and neuronal MCT2 (358). This idea has been recently exemplified by adenovirus-mediated conditional knockout of neuronal MCT2 and astrocytic MCT4 in a rodent study. Conditional knockout of neuronal MCT2 and astrocytic MCT4 impair learning and memory performances in the hippocampus-dependent tasks (359). Importantly, lactate infusion in the cerebral ventricles restores spatial learning in MCT4 knockout mice, suggesting that lactate shuttling by the astrocytes is pivotal to the direct effect of lactate, which contributes to hippocampal-dependent learning (359). Notably, neuronal MCT2 knockout, but not astrocytic MCT4 knockout, reduces the immature neuron population in the hippocampal dentate gyrus (359), suggesting MCT2 is involved in regulating adult neurogenesis in the hippocampus. All in all, these results have supported the lactate shuttle hypothesis and how astroglial-neuron shuttle is involved in hippocampal-dependent learning and memory.”

Please read carefully through the manuscript. There are some “the” missing here and there and a few extra spaces. Other than that, I did not find many misspellings (some are pointed out in my detailed comments).

SPECIFIC COMMENTS (MINOR)

P1, L34: I believe the authors mean “peripheral organs” instead of “systemic organs”? This phrase is used several places in the manuscript.

Response: Thank you for your reminder. We have changed to peripheral organs

P2, L57: The authors write: “Long-term potentiation (LTP) and long-term depression (LTD) are cellular mechanisms that indicate changes in synapse plasticity” Do the authors actually mean changes in synapse plasticity or rather that LTP and LTD are mechanisms involved in synaptic plasticity?

Responses: We have clarified it as Long-term potentiation (LTP) and long-term depression (LTD) are cellular mechanisms that indicate changes in synaptic plasticity (Line 90 - 91) as below: “Long-term potentiation (LTP) and long-term depression (LTD) are cellular mechanisms underlying changes in synaptic plasticity (69) ….”

P2, L78-79: The authors state that a reduction in the expression levels of PSD and gephyrin suggests that HFD affects both excitatory and inhibitory neurotransmitters in the hippocampus. A reduction in scaffolding proteins in the postsynaptic membrane is like not affecting the neurotransmitters per se (at least not directly), hence a more correct choice of words would perhaps be “excitatory and inhibitory synapses”.

Response: We have amended the expression to “excitatory and inhibitory synapses” accordingly (Line 86-89) as below: “Of note, HFD also reduces the expression levels of PSD-95 (a postsynaptic scaffolding protein in glutamatergic synapses) and gephyrin (a postsynaptic scaffolding protein mediating aggregation of GABAA receptors), suggesting that HFD could impair synaptic plasticity by modulating both glumatergic and GABAergic function in the hippocampus (87).”

P4. L146: I suggest rewriting the sentence “These macrophages can take up to 40% of all cells in the adipose tissue”. The macrophages certainly don not take up 40 % of the adipocytes, as this sentence implies.  What the authors mean is probably that the macrophages constitute 40 % of the cells (as opposed to 10 % in the normal condition)?

Response: We have re-written and clarified the contents with Weisberg et al study (2003) included as below:

Line 235 – 236: “During extreme obesity, it is estimated that these macrophages can take up over 50% of all cells in the adipose tissue (175)”

P5, L198: I do not understand the sentence “These perturbations further promote microglial activation as impair hippocampal plasticity”. Please revise.

Response: We have re-written the sentence as below:

Line 288 – 289: “These perturbations not only affect the peripheral metabolism, but they also trigger microglial activation in the hippocampus, which then impair hippocampal plasticity.”

P5, L199-200: Please clarify what is meant by “Studies have shown that dietary withdrawal is an ineffective non-pharmacological intervention for improving learning and memory deficits (96, 97)”. What do the authors mean by “dietary withdrawal”? The two sited papers demonstrate effects of dietary interventions, and the next couple of sentences also suggest that the authors mean that there is an effect of diets on measures of learning and memory.

Response: We have corrected the sentences as below:

Line 290 – 291: Studies have shown that withdrawal from high-fat diet is an effective non-pharmacological intervention for improving learning and memory deficits.

P7. L267: Stating that adipokines mediate (all of) the pro-cognitive effects of physical exercise seems like an over-simplification. What about myokines? The authors also present some factors other than adipokines that affect cognition. Hence, the above-mentioned sentence should be revised.

Response: We have improved the sentences as below:

Line 591 - 593: “Adipose tissue secrets IL-1β that can trigger the neuroinflammatory response in obesity- and diabetes, it also secrets endocrine hormones such as leptin (138) and adiponectin (59, 131) to mediate pro-cognitive effects of physical exercise.”

All other comments:

Response: Thank you so much for reviewer’s constructive comments. We have made substantial changes, corrected typos and clarified contents accordingly. We hope the changes are satisfactory to the reviewer.

Reviewer 2 Report

Comments to the Authors:

General Comments:

For the most part, this is a nicely written short review article that focus on how systemic metabolic disorders (such as obesity and diabetes) can be detrimental to brain health, plasticity, and cognition, offering a specific emphasis on the role of exerkines. This is a relatively new and exciting field of research and therefore this targeted review is timely and fulfils a current gap in the literature. At this moment, I only have a few minor comments that can be easily addressed before this manuscript can be accepted for publication (see detailed Specific Comments below).

Specific Comments:

1. Language: Although this is a very well written review and the quality of the language is quite good, there are a few grammatical mistakes throughout the text that should be corrected. Therefore, I encourage the Authors to carefully proofread the final version of the manuscript, paying particular attention to grammatical mistakes.

Introduction:

2. Lines 38-39: The authors refer to "metabolic syndrome". This term (and its clinical criteria) should be provided in the text before this term is used.

3. Lines 45-46: When explaining that hippocampal plasticity is involved in cognition (learning and memory) and affective behaviours, it would be appropriate to briefly explain the functional differences between the dorsal and ventral hippocampus (and cite appropriate references).

Section 3:

4. Lines 220-221: This sentence is not completely accurate. Oxidative Phosphorylation is part of the normal mitochondrial function. However, an increase in ROS production will result in disturbances/alterations of Oxidative Phosphorylation (and not in OXPHOS per se). This should be corrected.

Section 4:

5. Sub-sections 2.1 and 2.2: These sections are incorrectly numbered. They should be Sections 4.1 and 4.2.

6. Sub-section 2.2. Irisin: The recent study on the role of irisin in the anti-depressant and pro-neurogenic effects of physicial exercise by Siteneski et al. (2020 -- J. Neural Transm 127:355-370), should also be discussed and cited in this Section.

7. Conclusions: I believe including a summary Figure/Diagram that can convey the main points reviewed in this manuscript (e.g., how exerkines modulate the effects of physical exercise on hippocampal plasticity) would greatly improve its readability.

Author Response

Reviewer 2:

We thank you so much for reviewer’s kind comments. We have improved the manuscript with grammatical check, and added in new figure in conclusion according to reviewer’ comments.

Point to point response to major suggestions are listed as below:

  1. Lines 38-39: The authors refer to "metabolic syndrome". This term (and its clinical criteria) should be provided in the text before this term is used.

Response: We have provided a common description and clinical criteria on metabolic syndrome as below:

Line 33 - 36: “Metabolic syndrome encompasses a cluster of risk factors that could lead to cardiovascular diseases and diabetes (2). Criteria for clinical diagnosis of metabolic syndrome include increases in waist circumference, triglycerides, blood pressure, fasting glucose, and a reduction in high-density lipoprotein cholesterol (3)”

  1. Lines 45-46: When explaining that hippocampal plasticity is involved in cognition (learning and memory) and affective behaviours, it would be appropriate to briefly explain the functional differences between the dorsal and ventral hippocampus (and cite appropriate references).

Response: we have added in the relevant contents with citation

Line 77 - 80: “The hippocampus plays an essential role in regulating spatial learning and memory processes, as well as affective behaviours (46). Specifically, the dorsal hippocampus is involved in spatial learning and memory, while the ventral hippocampus is involved in mood regulation (47).”

  1. Lines 220-221: This sentence is not completely accurate. Oxidative Phosphorylation is part of the normal mitochondrial function. However, an increase in ROS production will result in disturbances/alterations of Oxidative Phosphorylation (and not in OXPHOS per se). This should be corrected.

Response: We have improved the accuracy of the contents accordingly as below:  

Line 322-324:  “Excessive ROS production can induce mitochondrial DNA damage, lipid peroxidation, as well as oxidative phosphorylation (OXPHOS) (202).”

  1. Sub-section 2.2. Irisin: The recent study on the role of irisin in the anti-depressant and pro-neurogenic effects of physicial exercise by Siteneski et al. (2020 -- J. Neural Transm 127:355-370), should also be discussed and cited in this Section.

Response: We have discussed and cited the suggested paper as below:

Line 658 – 661: “Another study investigating the role of irisin on mood regulation demonstrates that irisin is a mediator of exercise-induced antidepressant and neurotrophic effect, while the irisin mechanism of antidepressant action requires more in-depth investigations (337).”

Reviewer 3 Report

The review article titled "From obesity to hippocampal neurodegeneration: Pathogenesis and Therapeutics" by Suk Yu Yau and Thomas Hoi Yin Lee is of a relevant topic.

There are some articles authors can consider and include in the last paragraph of the introduction (page 4) before the section "2. From inflammation in adipose tissue to impaired hippocampus plasticity in obese and diabetic conditions". 

Can Diet and Physical activity limit AD risk. Rege SD et al.,Curr Alzheimer Res. 2017;14(1):76-93   The pancreas-brain axcis: insight into disrupted mechanisms associating type 2 diabetes and AD. Desai GS et al., J Alzheimers Dis. 2014;42(2):347-56. doi: 10.3233/JAD-140018.  

Also it will be good to add a schematic representation to explain how the systemic metabolic disorders could be detrimental to brain health and cognitive functions. It will be easy for readers to understand.

Author Response

Reviewer 3:

There are some articles authors can consider and include in the last paragraph of the introduction (page 4) before the section "2. From inflammation in adipose tissue to impaired hippocampus plasticity in obese and diabetic conditions".

Can Diet and Physical activity limit AD risk. Rege SD et al.,Curr Alzheimer Res. 2017;14(1):76-93   The pancreas-brain axcis: insight into disrupted mechanisms associating type 2 diabetes and AD. Desai GS et al., J Alzheimers Dis. 2014;42(2):347-56. doi: 10.3233/JAD-140018. 

Also it will be good to add a schematic representation to explain how the systemic metabolic disorders could be detrimental to brain health and cognitive functions. It will be easy for readers to understand.

Response: We have included Desai et al (Line 64-67) and Rege et al (Line 68-70) in the introduction and figure as suggested.

Line 64 – 67: “β-amyloid accumulation is one of the pathologies found in the AD brains. In the form of amylin or islet amyloid polypeptide accumulation, amyloidosis is also observed in the pancreas under diabetic condition (30).”

Line 68 – 70: “It has been previously reported that adipokines secreted from the adipose tissue could influence brain plasticity and cognitive function in both physiological and pathological conditions, respectively (32, 33).”

Also, we have added four schematics to demonstrate how diet and obesity can be linked up to neuroinflammatory response, synaptic deficits, and cognitive impairments, which can properly illustrate our ideas in this review

Reviewer 4 Report

The metabolic syndrome is one of the major risk factor, which include obesity and diabetes associated with insulin resistance, dysglycemia, and hypercholesterolemia, promote a variety of dangerous and costly chronic diseases that are among humanity’s most pressing public health problems. Although numerous genetic factors influence susceptibility to the development of metabolic syndrome, associated with increased incidence of this cognitive disorder has occurred amidst stark societal changes in food production and dietary habits has led to the general presumption that diet is a major determinant of metabolic syndrome. This Review manuscript from Thomas Lee and Suk-yu Yau focused on a dietary induced inflammation and its role in neurocognitive function under the condition of metabolic disorders obesity and diabetes. Based on the extensive ongoing research subject that is been discussed in this revised manuscript, it is acceptable for publication in Molecular Sciences. However, I have some minor comment and suggestion that could possibly discussed in this manuscript, which can enhance this review paper to bring more attention to readers.

  1. Since this review focused on the HFD (High Fat Diet) induced, obesity leads to pro-inflammatory response. Do you also think that total Lipids and Lipid droplets and any specific type of lipid speices play role in inducing the inflammatory response in the brain?
  1. Can you also discuss and comment briefly a one paragraph about the western diet induced inflammatory response in cognitive function.
  1. Please also discuss about the beneficial role of caloric restriction diet and intermittent fasting in the improvement of neurodegeneration and pathogenesis. If possible, also include the drugs or Calorie restriction mimetics and their beneficial role in the diet induced neuro inflammation.

Author Response

Reviewer 4:

Thank you so much for your encouraging comments.

Since this review focused on the HFD (High Fat Diet) induced, obesity leads to pro-inflammatory response. Do you also think that total Lipids and Lipid droplets and any specific type of lipid speices play role in inducing the inflammatory response in the brain?

Response: We have discussed how dietary lipids may have direct effect on neuroinflammatory response in the brain, regardless of their essential metabolic and physiological roles, in the new Section 5. Potential direct effect from dietary fat towards neuroinflammatory response

Line 380 – 397:

“5. Excessive palmitate consumption from diet triggers a direct neuroinflammatory response in the hippocampus

Palmitate is the most abundant saturated fatty acid present in the circulation (221) and cerebrospinal fluid (222). Increased brain uptake and accumulation of palmitate is reported in individuals with obesity and metabolic syndromes (223). Moreover, palmitate is increased in the cerebrospinal fluid of overweight and obese humans (224). Studies have suggested that palmitate impairs synaptic plasticity by elevated microglial activity. Palmitates suppress LTP formation in the Schaffer-collateral path (224). Local infusion of palmitate in the cerebral ventricles impairs learning and memory performance in the object recognition task, object location task, step-down task, and the Barnes maze task (224). In vitro study suggests that IRS-1 signaling is suppressed in the hippocampal neurons by the microglial-derived TNF-α (224), implicating palmitate-induced microglial activities may reduce insulin sensitivity in the hippocampal neurons. In another study, exosome fraction, which is isolated from palmitate-stimulated microglia in vitro, induces an immature dendritic spine phenotype in primary hippocampal neurons (225). Lastly, high-fat diet also induces similar predominance of immature dendritic spines from CA1 neurons alongside with diminished levels of the scaffold protein Shank2 and impaired spatial memory performance (225). These studies highlight the microglia-neuronal communication through exosomes, where palmitate-induced microglial inflammation may adversely influence on spine growth in the neighboring neurons.”

  1. Can you also discuss and comment briefly a one paragraph about the western diet induced inflammatory response in cognitive function.

Response: Thank you for your suggestion. We have included a discussion on western diet, inflammatory response and cognitive function (Line 28 – 57).

“Line 28 – 57: Western dietary pattern and sedentary lifestyle have fueled the obesity epidemic (1). Processed and refined food constitutes of saturated fats, added sugar, and salts, which contribute to high caloric intake under chronic consumption. Besides, the primary consumption of red meats and dairy products with a lack of vegetables and fresh fruits are the other characteristics of the Western dietary pattern. However, when caloric consumption far exceeds expenditure under prolonged physical inactivity, metabolic syndromes are developed. Metabolic syndrome encompasses a cluster of risk factors that could lead to cardiovascular diseases and diabetes (2). Criteria for clinical diagnosis of metabolic syndrome include increases in waist circumference, triglycerides, blood pressure, fasting glucose, and a reduction in high-density lipoprotein cholesterol (3). A recent review has summarized that the key pro-inflammatory constituents from the dietary sources, including saturated fatty acids, cholesterol, added sugars, refined grains, purines, dietary carnitine, and dietary histidine (4). Chronic and excessive dietary intake of these constituents may trigger a chronic inflammatory response in multiple tissue organs, which then develop into non-communicable diseases (4). Both diabetes and dementia are examples of non-communicable diseases. It is well-known that the Western dietary pattern is strongly linked to the development of obesity and Type 2 diabetes mellitus (T2DM) (5). Other studies also report the association between adopting a Western dietary pattern and impairments in hippocampal-dependent learning and memory performance across the lifespan (6-12). A clinical study further demonstrates that a causative relationship by subjecting a group of healthy young adults (n = 102) to a short exposure (four days) to a high caloric diet. Healthy young adults are subjected to high saturated fat and added sugar breakfasts (53% total fats, 37.5% carbohydrates, 11.5% proteins) or control (15.9% total fat, 31.8% carbohydrates, 51.3% proteins) for four days (13). Four days after, individuals consuming a high caloric diet have lower retention scores in the Hopkins-Verbal Learning Test as compared to the controls, implicating a decline in hippocampal-dependent learning and memory performance. Of particular note, a negative correlation between retention score and blood glucose level is observed. Moreover, interoceptive sensitivity to satiety is also reduced in these individuals consuming high caloric diet, suggesting a lower appetitive control.  Another clinical study further demonstrates a correlation between impaired appetitive control and decline in memory retention test score after one week of high caloric diet consumption (n = 110) (14). The enigmatic relationship underlying high-caloric diet, obesity, diabetes, and hippocampal plasticity has been actively investigated in rodent studies.”

  1. Please also discuss about the beneficial role of caloric restriction diet and intermittent fasting in the improvement of neurodegeneration and pathogenesis. If possible, also include the drugs or Calorie restriction mimetics and their beneficial role in the diet induced neuro inflammation.

Response: We have included a new section 8 to discuss about the pro-cognitive and neurotrophic effects of caloric restriction and intermittent fast, as well as its underlying mechanisms

Line 551 – 588:

“8. SIRT1/SIRT3 axis potentiates the pro-cognitive effects of caloric restriction and intermittent fasting

Sirtuins (SIRT) are NAD+-dependent deacetylases (271). SIRT1 is a cytosolic and nuclear sirtuin modulates transcription factors through histone deacetylation (265, 272-275). SIRT3 is a mitochondrial sirtuin regulating β-oxidation (276). SIRT1/SIRT3 axis aims to promote energy storage intracellularly and to resist oxidative stress (277, 278).

In vivo studies illustrate that SIRT1 is a mediator of caloric restriction-induced mitochondrial biogenesis in mice (279-281), while SIRT1 knockout abrogates the metabolic benefits of caloric restriction (274, 282). In the hippocampus, 23 weeks of HFD reduces the expressions of Ppargc1a, Ppp1cb, Reln and Sirt1 (283). Increased DNA methylation and decreased DNA hydroxymethylation are observed in the promoter region of Sirt1, implicating a diet-induced epigenetic modulation of Sirt1 (283). On the contrary, caloric restriction promotes learning and memory performance as well as hippocampal synaptic plasticity by upregulating signaling transducers involved in mitochondrial biogenesis (284), including CREB phosphorylation and increased expression of SIRT1, PGC-1α, nitric oxide synthase, and phosphoenolpyruvate carboxykinase in the hippocampus (284). The activation of a mitochondrial biogenesis signaling pathway is associated with better learning and memory performance in the novel object recognition task and enhanced LTP formation in the Schaffer-collateral path in a neuronal-specific CREB-dependent manner (284).

SIRT3 is one of the SIRT1 deacetylation targets. HFD-induced obesity and aging obliterate SIRT1-mediated deacetylation of SIRT3, which reduces SIRT3 activity and stability in the mouse model (285). Intermittent caloric restriction, as short as one week to one month, promotes the hippocampal SIRT3 expression (286). Besides, the anxiolytic and pro-cognitive effects of intermittent caloric restriction are abolished when SIRT3 is selectively knockout in the hippocampal neurons. Intermittent caloric restriction further impairs LTP formation in the Schaffer-collateral path of the SIRT3-deficient mice. Significantly, intermittent fasting reduces seizure incidence, restores hippocampal-dependent spatial memory performance, as well as rescues LTP deficits in AppNL-G-F mice of AD.

SIRT3 can deacetylate LKB1, resulting in AMPK activation and mitochondrial biogenesis (287). Four-week intermittent fasting activates AMPK/PGC-1a/mTORc/COX4/ND-1 signaling cascade, suggesting that intermittent fasting promotes mitochondrial biogenesis in the hippocampus. The increased mitochondrial DNA content further evidences this result in the hippocampus in db/db mice (288). Essentially, intermittent fasting also exerts pro-cognitive and neurotrophic effects by restoring diabetes-induced hippocampal-dependent spatial memory deficits and the morphology of postsynaptic density of the neurons in the hippocampal CA1 region in addition to the activation of BDNF/pCREB/PSD-95 signaling cascade (288).

In sum, SIRT1/SIRT3 axis is one of the molecular targets of caloric restriction-induced pro-cognitive and neurotrophic effects, partly due to its functional role in regulating mitochondrial biogenesis.”

Round 2

Reviewer 3 Report

Authors have answered all the comments raised and it has improved the quality of the manuscript.